# Architecture of the yeast Elongator complex

Maria I Dauden[1], Jan Kosinski[1], Olga Kolaj-Robin[2,3,4], Ambroise Desfosses[1,†], Alessandro Ori[1,‡], Celine Faux[2,3,4,§], Niklas A Hoffmann[1], Osita F Onuma[5], Karin D Breunig[5] iD, Martin Beck[1] iD, Carsten Sachse[1] iD, Bertrand Séraphin[2,3,4], Sebastian Glatt[6,*] & Christoph W Müller[1,**] iD

## Abstract

The highly conserved eukaryotic Elongator complex performs specific chemical modifications on wobble base uridines of tRNAs, which are essential for proteome stability and homeostasis. The complex is formed by six individual subunits (Elp1-6) that are all equally important for its tRNA modification activity. However, its overall architecture and the detailed reaction mechanism remain elusive. Here, we report the structures of the fully assembled yeast Elongator and the Elp123 sub-complex solved by an integrative structure determination approach showing that two copies of the Elp1, Elp2, and Elp3 subunits form a two-lobed scaffold, which binds Elp456 asymmetrically. Our topological models are consistent with previous studies on individual subunits and further validated by complementary biochemical analyses. Our study provides a structural framework on how the tRNA modification activity is carried out by Elongator.

**Keywords** electron microscopy; Elongator; *Saccharomyces cerevisiae*; tRNA modification; yeast
**Subject Categories** RNA Biology; Structural Biology

See also: **DT Setiaputra** *et al*

## Introduction

During the elongation phase of the ribosome-mediated translation process, transient pausing events support proper domain folding of the nascent polypeptide chains, which gain their three-dimensional conformations immediately after they have left the exit tunnel of the ribosomes, a process sometimes facilitated by chaperones [1–3]. Previous studies indicated that specific base modifications in the wobble base position of tRNAs are crucial to maintain these highly dynamic and complex mechanisms. Mainly because they influence the recognition rate and affinity between incoming tRNAs and codons in the A site of the translating ribosome [4–6]. Interestingly, expansion of tRNA gene copy numbers and isoacceptors correlate well with an increase in different tRNA modification enzymes, suggesting an evolutionary selection for optimizing translational efficiency and accuracy via different modification mechanisms [7].

The eukaryotic Elongator complex has been associated with a plethora of cellular activities [8–10], but nowadays, it is widely accepted that the main cellular function of the Elongator complex is the formation of 5-methoxycarbonylmethyl-uridine (mcm$^5$U), 5-methoxycarbonylmethyl-2-thio-uridine (mcm$^5$s$^2$U), and 5-carbamoylmethyl-uridine (ncm$^5$U) in the wobble base position of 11 eukaryotic tRNAs [10]. Nevertheless, the detailed chemistry of the Elongator modification reaction is insufficiently described, and the role of the resulting modifications is not fully understood [11–13]. In particular, it is currently unclear how tRNA is delivered to the active center and how the high modification turnover can be achieved in the context of the full complex. The cellular role of Elongator is of fundamental clinical relevance, as mutations affecting the integrity and activity of this macromolecular complex are related to the onset of neurodegenerative diseases [14–16], cancer [17,18], and intellectual disabilities [19].

The fully assembled Elongator complex contains two copies of each of its six subunits *in vivo* [20] and has an estimated molecular weight of ~850 kDa. All subunits are highly conserved among eukaryotes [21], which has also been experimentally proven by cross species complementation analyses of genes encoding individual subunits and subdomains for yeast, insects, worms, plants, and humans [15,22–26]. Shortly after the initial description of the three-component Elongator sub-complex (Elp123) [27], an additionally associated sub-complex containing subunits Elp4, Elp5, and Elp6

1    European Molecular Biology Laboratory, Structural and Computational Biology Unit, Heidelberg, Germany
2    Université de Strasbourg, IGBMC, Illkirch, France
3    CNRS, IGBMC UMR 7104, Illkirch, France
4    Inserm, IGBMC U964, Illkirch, France
5    Institute of Biology, Martin Luther University Halle-Wittenberg, Halle (Saale), Germany
6    Max Planck Research Group at the Malopolska Centre of Biotechnology, Jagiellonian University, Krakow, Poland
      *Corresponding author. Tel: +48 12 664 6321; E-mail: sebastian.glatt@uj.edu.pl
      **Corresponding author. Tel: +49 6221 387 8320; E-mail: cmueller@embl.de
      †Present address: School of Biological Sciences, University of Auckland, Auckland, New Zealand
      ‡Present address: Leibniz Institute on Aging-Fritz Lipmann Institute, Jena, Germany
      §Present address: CRBM − CNRS UMR5237, Montpellier, France

    

[28] was identified under milder purification conditions. Crystal structures are currently available for the homo-dimer of the Elp1 C-terminus [29], Elp2 [30], and the RecA-like Elp456 hetero-hexamer [20,31]. We recently determined the crystal structure of full-length Elp3 from a bacterial homolog (DmcElp3), which shows high sequence similarity to Elp3s from different organisms, including yeast and humans [32]. The structure shows that the lysine acetyl transferase (KAT) domain and the S-adenosylmethionine binding domain (SAM) share a large and highly conserved interface that creates a specific tRNA binding pocket and forms a composite active site. However, no structural information is yet available for the Elp123 sub-complex or the holoElongator (Elp1-6), precluding the localization of the active center in relation to the other complex components.

We set out to obtain structural information on the fully assembled Elongator complex to better understand how the different subunits interact and together deliver modifiable tRNAs to the enzymatically active Elp3 subunit. Here, we report the structures of holoElongator and the Elp123 sub-complex at 31 Å and 27 Å resolution, respectively, by negative-stain electron microscopy (EM). In addition, we describe the global interaction network between all six individual subunits using crosslinking mass spectrometry (XL-MS) and combine these results by an integrative modeling approach, which allows us to localize all subunits and provide a topological model of the full complex. The model enables us to anticipate how the tRNA modification activity is carried out by this large molecular machine and how the individual subunits contribute to its assembly and activity.

# Results

## Endogenous Elongator and Elp123 sub-complex coexist as stable complexes

In an initial attempt of assembling the Elongator complex from individually purified proteins, expressed heterologously in *Escherichia coli*, we were able to observe a direct interaction between Elp1 and Elp3, and Elp1 and Elp456. Furthermore, these five subunits can also simultaneously interact with each other to form a complex that comprises Elp1, Elp3, and the Elp456 sub-complex. In contrast, we could not detect interactions between Elp2 and any of the other subunits (Fig EV1A and B), supporting the idea that further posttranslational modifications or chaperones are required for the assembly of the complete Elongator complex [33,34]. Therefore, we focused on the characterization of endogenous tandem-affinity purification (TAP)-tagged [35] Elongator, which permits the purification of all six subunits directly from yeast. We constructed yeast strains carrying endogenously TAP-tagged versions of Elp1 and Elp6 and purified Elongator using previously established purification protocols [20]. Consistent with sub-stoichiometric cellular amounts of Elp456 *in vivo* [28,36,37], purifications of Elp1-TAP resulted in an excess of Elp123 sub-complex [20], whereas Elp6-TAP purifications resulted in reduced quantities but yielded highly pure, complete, and stoichiometric Elongator complex. Large-scale preparations yielded sufficient amounts of pure Elongator complex and Elp123 sub-complex to analyze their overall architecture and shape by EM.

## The Elp456 ring is asymmetrically positioned in the holoElongator complex

In order to obtain the structure of Elongator, we used negative-stain EM of the purified Elp6-TAP holoElongator complex. Highly pure and stoichiometric holoElongator complex (Fig 1A) was stabilized using low amounts of glutaraldehyde and reapplied to gel filtration. A homogeneous population of two-lobed particles was observed (Fig 1B) and 22,876 particles were selected and subjected to reference-free two-dimensional (2D) classification. The class averages show a complex of around 260 × 170 Å formed by two lobes with a cleft in between (Fig 1C). Interestingly, a hexameric ring-shaped density protruding from one of the lobes is clearly visible (Fig 1C). In addition, some of the side view 2D averages are asymmetric, while some top views do not show a ring. Together, these observations suggest that only one copy of the hetero-hexameric Elp456 ring asymmetrically interacts with the Elp123 sub-complex, as previously described [20].

The structure of holoElongator (Fig 1D) is ~270 Å in height, ~180 Å length, and ~160 Å width, and there is a clear correlation between the class averages and the model back-projections (Fig 1C). The Elongator model shows an estimated resolution of 31 Å based on the Fourier Shell Correlation (FSC; Fig EV2A), and although the angular assignment of particle orientations is well distributed, we observed a preferential orientation, a commonly observed phenomenon in negative-stain EM (Fig EV2B). The structure consists of two lobes linked in the upper part by the "arch" and divided by a "cleft", with two lateral bean-shaped densities, herein after referred as the "wings". A hexameric ring density stands out from one of the lobes and the Elp456 crystal structure seamlessly fits into this density (Fig 1E). Thus, although most of the Elongator density has the shape of a nearly symmetrical two-lobed structure, the Elp456 ring binds to the complex asymmetrically. The C-terminal domain (CTD) of Elp1 observed in the crystal structure [29] can be also unambiguously fitted to the holoElongator model (Fig 1F). The Elp1-CTD fits to the arch of the Elongator complex with higher scores than in other locations. Importantly, the fit of Elp1-CTD indicated the correct handedness of the 3D reconstruction as the Elp1-CTD fits significantly better in one of the mirrored holoElongator maps (Fig EV2C and D). In addition, we used an experimental tilt-pair validation approach to confirm the handedness and correctness of our 3D reconstruction (Fig EV2E). However, we could not unambiguously place either the WD40 domains of Elp1 or the Elp2 and Elp3 subunits, which fitted to various locations in the map with similar cross-correlation scores, likely due to the limited resolution of our reconstruction.

## Two copies of the Elp1, Elp2, and Elp3 subunits form a symmetric Elp123 sub-complex

To identify and localize the Elp1, Elp2, and Elp3 subunits, we analyzed purified Elp123 sub-complex by negative-stain EM. As previously described [20,28], Elp1-TAP purifications contain a heterogeneous mixture of Elp123 and holoElongator, which makes the structural analyses of these samples more complicated. Nevertheless, after slight modification of the purification protocol, we managed to separate the two species and obtain sufficient amounts of pure and stoichiometric Elp123 sub-complex. The purified Elp123

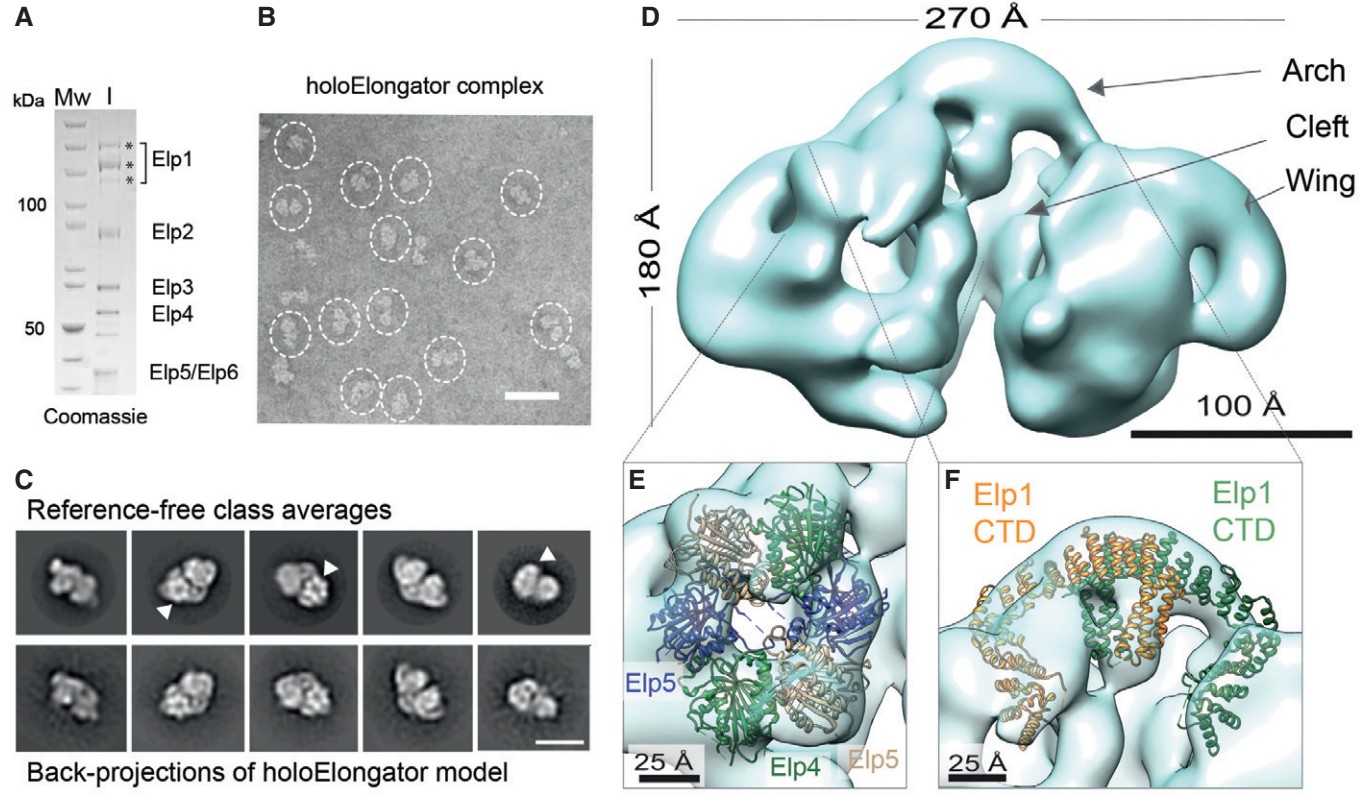

**Figure 1. EM reconstruction of endogenous holoElongator.**

A  SDS–PAGE gel showing the purified holoElongator complex used as input for gel filtration. Protein bands marked with asterisks presumably correspond to different phosphorylation states of Elp1 that have been previously described [34].

B  Representative negative-stain EM field of the holoElongator complex. Particles in side and tilted views are highlighted. Scale bar, 50 nm.

C  Reference-free class averages and back-projections of the holoElongator model. Hexameric ring-shaped densities and the asymmetric side view are indicated (arrowheads). Scale bar, 20 nm.

D  EM reconstruction of the holoElongator complex at 31 Å resolution.

E  Fitting of the Elp456 crystal structure into the ring density of the holoElongator reconstruction.

F  Fitting of the Elp1 CTD into the holoElongator reconstruction.

sample (Fig 2A) was stabilized with low amounts of glutaraldehyde and eluted from a gel filtration column at a volume expected for a ~610 kDa complex. This suggested that the Elp123 sub-complex in isolation also harbors two copies of each of the three subunits, as previously described for the full complex [20,28].

A major population of two-lobed particles was observed (Fig 2B) and 50,034 particles were selected for reference-free 2D classification. The class averages show different side views of a structure that clearly resembles holoElongator but misses the ring-shaped density (Fig 2C). 3D classification of the Elp123 sub-complex showed the presence of a twofold symmetry axis, which was previously observed in some of the class averages (Fig EV3A). Thus, the final C2-symmetrized model shows an estimated resolution of 27 Å according to the FSC (Fig EV3B and C). The back-projections of the 3D model correlate well with the class averages (Fig 2C), and the overall angular assignment of particle orientations is well distributed (Fig EV3D).

The 3D reconstruction of the Elp123 complex closely resembles the holocomplex. In detail, the two lobes are linked in the upper part by the arch and are separated in the lower part by the cleft, while they extend laterally into the wings (Fig 2D). The dimensions

of the Elp123 sub-complex are also similar in height (~170 Å) and length (~260 Å) to the holocomplex, though it is more flattened in the third dimension (~100 Å) due to the absence of Elp456. Similarly to the holoElongator, the dimer of the C-terminal domain (CTD) of Elp1 could be fitted significantly better in one of the mirrored Elp123 maps confirming the correct handedness of the Elp123 reconstruction (Figs 2E and EV4A). The Elp1-CTD fits to the arch in the equivalent position and the twofold symmetry axis of the Elp1-CTD structure aligns with the symmetry axis of the map.

Interestingly, some of the class averages in top view seemed to represent only partial Elp123 sub-complexes without one of the wings (Fig EV4B). Indeed, 3D classification showed one class that clearly resembled the intact Elp123 model missing one of the wings (Fig EV4C). The refinement of this class yielded an asymmetric model at 31 Å resolution based on the FSC (Fig EV4D). Back-projections of the 3D model correspond well with the class averages (Fig EV4B). A difference map between the full Elp123 model and the asymmetric model rendered, apart from a minor difference in the bottom part, a bean-shaped density (Fig EV4E). Our group also independently solved the yeast Elp2 structure at 2.8 Å resolution (Appendix Table S1 and Fig EV4F), which is highly similar

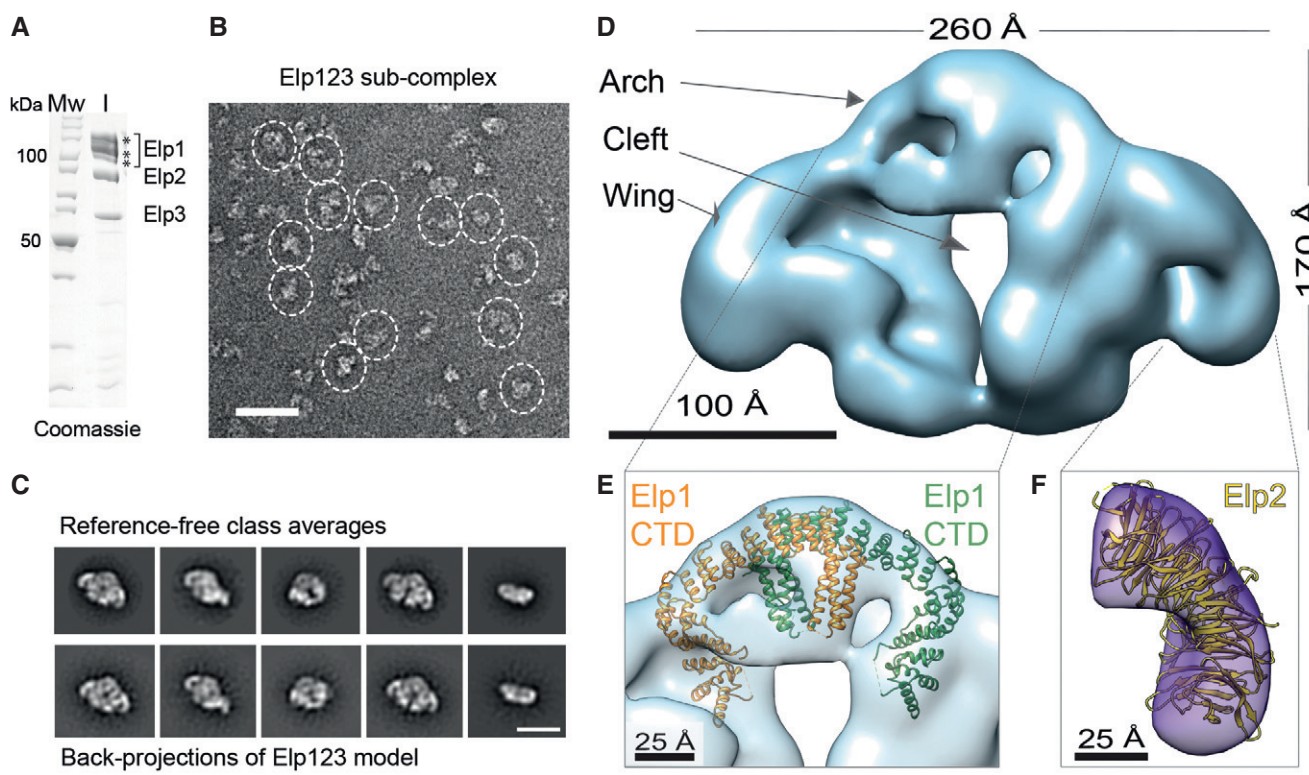

**Figure 2. EM reconstruction of endogenous Elp123 sub-complex.**

A   SDS–PAGE gel showing the purified Elp123 sub-complex used as input for gel filtration. Protein bands marked with asterisks presumably correspond to different phosphorylation states of Elp1.

B   Representative negative-stain EM field of the Elp123 sub-complex. Particles in side and top views are highlighted. Scale bar, 50 nm.

C   Reference-free class averages and back-projections of the Elp123 model. Scale bar, 20 nm.

D   EM reconstruction of the Elp123 sub-complex at 27 Å resolution.

E   Fitting of the Elp1 CTD into the Elp123 reconstruction.

F   Fitting of the Elp2 crystal structure in the difference density generated by subtracting the partial Elp123 reconstruction from the complete Elp123 reconstruction.

(r.m.s.d. = 1.19 $Å_{686Cα}$) to the previously published Elp2 structure [30]. Both Elp2 structures can be seamlessly fitted in the wings of the Elongator complex (Fig 2F), but due to the pseudo symmetric shape of its two WD40 domains it is not possible to know the exact orientation of the Elp2 at the attained resolution.

In summary, the presented 3D models allow us to position the CTD of Elp1 within the arch and the Elp2 within the wings, though the precise orientation of Elp2 is still unclear. Moreover, the Elp456 binds asymmetrically to one of the lobes formed by single copies of Elp1, Elp2, and Elp3. In contrast, the two N-terminal WD40 domains of Elp1 and the Elp3 subunit fit in different locations with similar scores, precluding their placement solely based on the EM maps and requiring additional restraints for their unambiguous localization.

**A tight network of interactions connects the Elp123 and the Elp456 sub-complexes**

In order to identify contacts between all the individual subunits and obtain additional spatial restraints, we performed XL-MS analyses (Fig 3 and Appendix Fig S1A). In summary, we detected 41 unique inter-subunits and 75 intra-subunit highly confident crosslinks. The crosslinks were deemed highly confident if their ld (linear discriminant) confidence score, as calculated by xProphet [38], was ≥ 30 (Appendix Table S2). Although many crosslinks appear in flexible regions, we were able to validate our approach by observing the expected distance (< 30 Å) between several lysine residue pairs present in the structured parts of the previously published crystal structures of Elp2, Elp3, and Elp456 and that had been identified as high confidence crosslinks (Appendix Fig S1B and C). Although the used Elp1-TAP sample contained an excess of Elp123, we were able to detect a tight network of interactions between the Elp123 and Elp456 sub-complexes. Importantly, Elp3 and Elp4 extensively crosslink to both the N- and C-terminal regions of Elp1 (Fig 3). Moreover, the exclusive presence of inter-subunit crosslinks on one of the sides of the Elp456 hetero-hexameric ring clearly identified a preferential orientation of the Elp456 ring on the Elp123 sub-complex (Appendix Fig S1D). Comparing to Elp1, Elp3, and Elp4, we observed only very few inter-subunit crosslinks in Elp2, Elp5, and Elp6. This may result from the lower level of disordered regions in Elp2, Elp5, and Elp6, but also may suggest that these subunits simply share less and smaller interfaces with the other subunits. In summary, we detected a unique set of distance restraints between the six Elongator subunits.

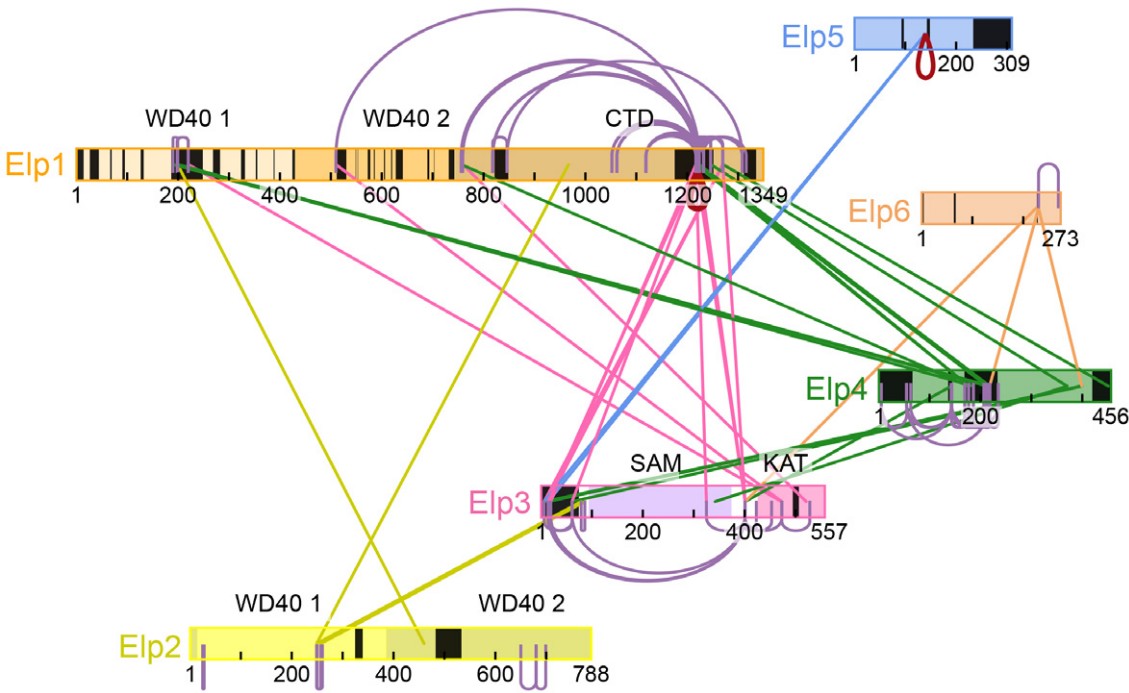

**Figure 3. XL-MS analyses of fully assembled Elongator.**
Schematic overview of highly confident (LD score at least 30) crosslinks between Elp1 (orange), Elp2 (yellow), Elp3 (pink), Elp4 (green), Elp5 (blue), and Elp6 (sand). The size of the used squares is adjusted to the length of the individual proteins and also indicated the location of the respective crosslinks. Inter-crosslinks are drawn as lines colored according to one of the linked subunits, intra-crosslinks as arcs colored according the subunit, whereas "dimeric" crosslinks (linking same residues of a protein) are depicted as red loops. Structural domains are indicated. Protein regions not present in the crystal structures and homology models are colored black. Figure was drawn using xiNET [76].

## The Elp1 dimer forms a scaffold for the Elp2 and Elp3 binding in the Elp123 sub-complex

Since the EM map of Elp123 sub-complex did not allow the unambiguous fitting of all Elongator components, we used a previously established integrative approach for modeling based on the combination of EM maps and crosslinks [39,40]. This approach performs global fitting of subunits or their domains in EM maps to maximize their fit to both the EM density and the crosslinks, optimize connectivity between domains of the same protein, and minimize the steric overlap between the domains.

As starting structures of the subunits, we used homology models of the WD40 domains of Elp1, a composite model of the Elp1-CTD dimer based on the yeast and human Elp1-CTD structures, the Elp2 crystal structure, and a homology model of Elp3 based on DmcElp3. Short loops and tails that were missing in the crystal structures or homology models, but formed crosslinks potentially useful for positioning the subunits were included as flexible loops and α-helices in accordance with secondary structure predictions. Although the exact conformation of such loops is difficult to predict, they can help in restraining the orientation of structured domains [41]. Overall, 30 out of 116 crosslinks could be mapped to the respective starting structures.

The best-scoring model fits to the Elp123 negative-stain EM map, fills 93% of the map volume, and recapitulates its characteristic shape (Fig 4A). Twenty-eight out of 30 (93%) crosslinks satisfy the distance threshold between crosslinked Cα atoms of 30 Å. The two

crosslinks that remain violated link the K401 residue from a helix located in a long loop not present in the Elp3 crystal structure with distant regions of Elp3 and Elp1-CTD. Thus, these crosslinks can be explained either by structural flexibility or the expected 5% false-positive rate of the XL-MS. The satisfied crosslinks define the locations and orientation of the subunits. The location as well as the orientation of Elp3 is defined by crosslinks from the N-terminus of Elp3 to Elp2 and from the Elp3-KAT domain to the second WD40 domain of Elp1 and Elp1-CTD (Fig 4B). The orientation of Elp2 in the wing of the map is restrained by crosslinks from the N-terminal WD40 domain of Elp2 to Elp1-CTD and the N-terminus of Elp3 (Fig 4B).

In summary, the model reveals the overall architecture of the Elp123 sub-complex of Elongator. The scaffold of the structure is formed by Elp1, which assumes the central position in the complex and mediates the dimerization. The Elp1-CTD dimer forms the arch and with its N-terminal WD40 domains creates a docking platform for Elp3. Elp2 is located peripherally attached to Elp1-CTD and Elp3. All subunits contribute to the formation of the central cavity, which consists of the space between the two lobes and extends toward the Elp2 wings.

### Topological model of the holoElongator

The above model of the Elp123 sub-complex is a suitable starting point for generating the model of the holoElongator. Thus, to build the complete model of the Elongator complex based on the EM

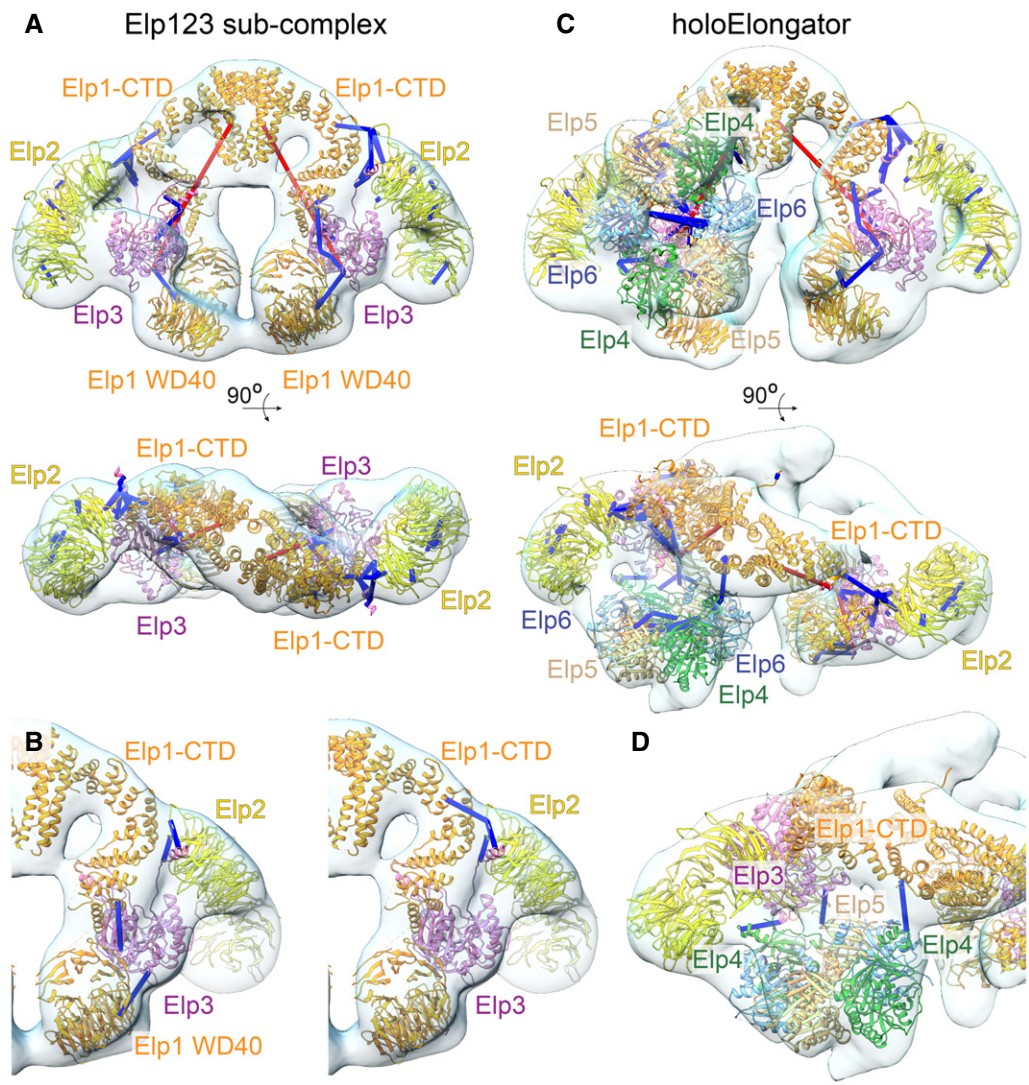

**Figure 4. Integrative models of the Elp123 sub-complex and the holoElongator complex.**

Models are shown in ribbon representation fitted to the corresponding EM maps. The lysine–lysine crosslinks are mapped on the models. The crosslinks are colored blue if they are satisfied (Cα–Cα atoms of the linked positions less than 30 Å apart) or red if violated. The crosslink visualization was created with the Xlink Analyzer plugin for UCSF Chimera [77].

A   The model of the Elp123 sub-complex.

B   Crosslinks supporting the orientation of Elp3 (left) and Elp2 (right). The crosslinks are shown separately to highlight crosslinks linking from Elp3 and Elp2 to other subunits, respectively.

C   The model of holoElongator.

D   Crosslinks supporting the orientation of the Elp456 hexamer.

map, we fitted our Elp123 structural model and the known Elp456 crystal structure. Because of the observed conformational deviations in the EM map of holoElongator, we fitted the two Elp123 lobes independently and refined the resulting model using an automated modeling procedure (see Materials and Methods). One Elp123 lobe fits well into the lobe bound to the hexamer, with all densities of Elp1, Elp2, and Elp3 recapitulated in the EM map. Thirty-seven out of 39 (95%) crosslinks satisfy the distance threshold of 30 Å between crosslinked Cα atoms. Elp2, Elp3, and part of the Elp1-CTD also fit well to the second lobe of the map. However, the fit is poorer for the WD40 domains of Elp1. The

Elp456 hetero-hexamer unambiguously fits to the hexameric density, while crosslinks provide information about the relative positions of the Elp4, Elp5, and Elp6 subunits within the hexameric ring and about the face of the ring that interacts with Elp123 (see below). Some loops and peripheral regions of Elp3 protrude out of the EM density but locate close to unassigned densities linking Elp456 and Elp3. Thus, Elp1 and Elp3 may undergo conformational changes upon binding of the Elp456 sub-complex.

The resulting model agrees with the crosslinks between the Elp456 and Elp123 sub-complexes (Fig 4C). Firstly, one of the copies of Elp4 locates close to Elp3, satisfying the crosslinks to Elp3.

The same Elp4 molecule is also close to the N-terminal WD40 of Elp1, likely explaining crosslinks to this domain (Fig 3). The second copy of Elp4 is placed close to the Elp1-CTD, enabling crosslinks from Elp4 to that region of Elp1 (Fig 4D). Clearly, crosslinks to both the CTD and the N-terminal WD40 domain of Elp1 can only be satisfied in the observed lateral ring position. The Elp456 positioning also explains the presence of crosslinks exclusively on one side of the ring (Appendix Fig S1D).

Our structural models suggest that the Elp456 hexamer is placed exactly on top of the cleft containing the Elp3 active site. The comparison of the maps also revealed that Elp123 lobes moved relatively to each other, at the hinge region in the middle of Elp1-CTD (Figs 1F and 2E). Thus, in addition to the presence of Elp456, the Elp123 lobes of holoElongator significantly deviate from the C2 symmetry observed in the Elp123 sub-complex. In summary, we provided topological models of the Elp123 subcomplex and holoElongator by integrative modeling, which are based on the combination of determined EM structures and XL-MS data.

### Functional *in vivo* and *in vitro* experiments validate the Elongator model

In order to validate the newly discovered subunit interfaces of these models, we performed biochemical, biophysical, and functional *in vivo* experiments. In detail, we used individually purified *S. cerevisiae* Elp1, Elp2, and Elp456 proteins and the KAT and SAM domains of Elp3 in different protein–protein interaction assays. Consistent with our crosslinking results and as predicted from our model, we observed that Elp1 interacts directly with the KAT domain of Elp3, but requires Elp456 to interact with the SAM domain of Elp3. In the presence of Elp1, Elp456 interacts with the SAM domain of Elp3, but not with its KAT domain (Fig 5A). We observed that $Elp45_{1-270}6$ directly interacts with the SAM domain of Elp3 as predicted, though a large excess of protein is required. In addition, we observed that in the presence of Elp1, the SAM domain of Elp3 and Elp456 seem to interact stronger, confirming the dense interaction network between Elp1, Elp3, and Elp4 in our XL-MS approach and the holoElongator model. However, N- and C-terminally truncated Elp456 ($Elp4_{66-426}5_{1-270}6$) [20] do neither interact with the SAM domain of Elp3 nor tether Elp1 to the SAM domain (Fig 5B).

Considering the rather surprising asymmetrical localization of Elp456, we were also interested in mapping the interacting regions between Elp123 and Elp456 more precisely. Firstly, a loop region in Elp4 (aa168–233) that contained a large number of inter-subunit crosslinks with Elp1 and intra-subunit crosslinks with the N-terminus of Elp4, is dispensable for the interaction between Elp1 and Elp4. Thus, we show that the highly conserved ten first residues of the Elp4 N-terminus, which also shows several intra-crosslinks to the above-mentioned loop (aa168–233), are important for the interaction between Elp1 and the Elp456 sub-complex (Fig 5C). Elp4 by itself interacts with the first WD40 domain of Elp1 (aa42–431; Appendix Fig S2A) and, accordingly, a deletion of the short conserved stretch at the N-terminus of Elp4 abolishes the interaction between $Elp1_{1-734}$ and Elp4 (Fig 5D). Moreover, we used isothermal titration calorimetry (ITC) to confirm that synthetic peptides (aa1–10 and aa1–27) of the conserved region directly interact with

the purified N-terminal region of Elp1 ($K_{d\ 1-10}$ = 2.7 μM ± 0.3, $K_{d\ 1-27}$ = 1.6 μM ± 0.4; Appendix Fig S2B). Interestingly, we observed that this interacting region between Elp1 and Elp456 is salt sensitive (Appendix Fig S2C), which has also been described for the interaction of the endogenous Elp123 and Elp456 sub-complexes [28]. Although the N-terminus of Elp4 is most likely unstructured and its length is not highly conserved, our model supports the presence of an interaction between the N-terminus of Elp4 and the N-terminal WD40 domain of Elp1.

Finally, to understand the contribution of individual conserved surface residues in Elp2, we mutated residues in the previously identified [30] basic region (R626, R628, R654, R675), which was described to be important for histone acetylation and microtubule binding. Mutating these residues also leads to phenotypes associated with tRNA modification defects. Furthermore, we also show that two additional regions (aa201–204 and aa552–557) seem to be important for Elongator activity as they show similar phenotypes (Fig EV5A and B). Accordingly, the former region (aa201–204) contributes to formation of the cavity around the active site, together with Elp1-CTD and Elp3, in which Elp456 binds. None of the tested mutations affects the stability of Elp2 (Fig EV5C) and we therefore believe that these regions are involved in protein–protein interactions.

## Discussion

Here, we determine the overall architecture of the endogenous eukaryotic Elongator complex and propose the relative position and orientation of the individual subunits within the fully assembled complex. Our study, together with a study published by Setiaputra *et al* [42] in the same issue, confirms that Elp456 asymmetrically interacts with the Elp123 sub-complex, to form holoElongator. Although the atomic positioning of Elp3 and the WD40s of Elp1 will require additional studies, the presented models suggest that Elp1 provides a scaffold for Elp2 and Elp3 and also acts as the docking platform for Elp456 (Fig 6A). In detail, the active site of Elp3 is located in the conserved cavity formed by Elp3, the N-terminal WD40 domain of Elp2, and the α-solenoid domain of Elp1-CTD (Appendix Fig S3A and B). The region of Elp2 contributing to the cavity includes residues 201–204, which were found to be important for tRNA modification activity of Elongator (Fig 6B). Furthermore, Elp3 is located close to the region of Elp1 implicated in tRNA binding (aa1221–1259) [43], suggesting that Elp3 and Elp1 might be able to simultaneously bind to a single tRNA molecule (Fig 6B).

Overall, our model agrees with the independently identified protein–protein interfaces between Elp1, Elp3, and Elp456 and with the relative orientation of the conserved regions of all individual subunits (Appendix Fig S3C–E). In detail, Elp3 is mostly anchored to Elp1 via its KAT domain, agreeing with our observations that Elp1 interacts with the KAT domain, but only shares a very small interface with the SAM domain of Elp3 (Fig 6C). Although the Elp456 hexamer is placed exactly on top of the cleft containing Elp3 active site, biochemical analyses suggest that the ring is also tethered to Elp123, via additional interactions, one of them between the flexible N-terminal residues of Elp4 and the first WD40 domain of Elp1 (Fig 6A). The position of Elp456 close to the active site of Elp3

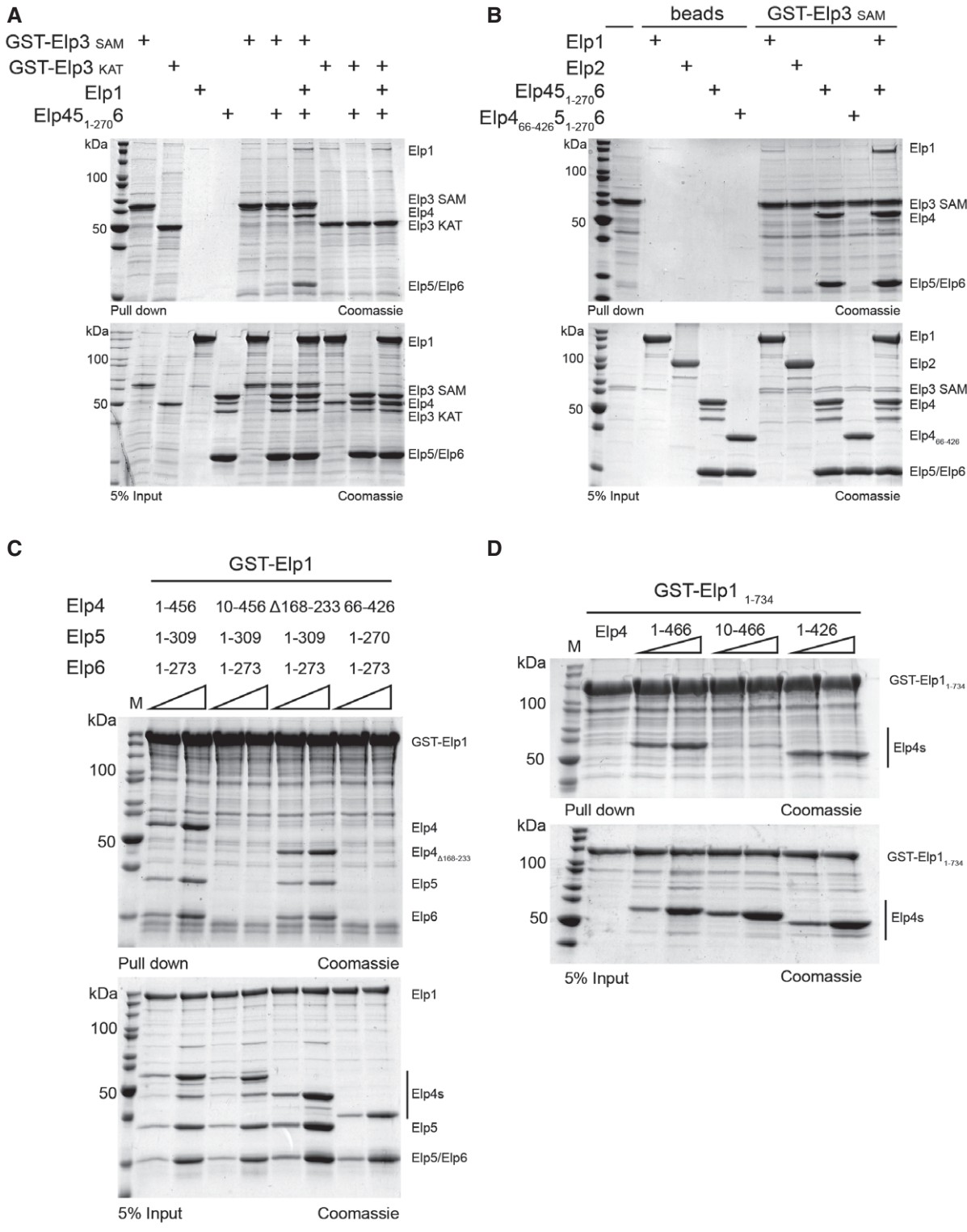

**Figure 5.  Validation of the Elp123 and holoElongator models.**

A  GST pull-down assays of purified GST-tagged Elp3 SAM domain (aa72–389) and KAT domain (aa348–557) with untagged full-length Elp1 and Elp45$_{1-270}$6. GSH resin and proteins were used as input controls. Lower panel shows 5% of the input and upper panel shows bound fractions. All samples were analyzed by SDS–PAGE and stained by Coomassie. Identities of respective proteins are indicated on the right.

B  Same as (A), but using only GST-tagged Elp3 SAM domain with combinations and variants of Elp1, Elp2, and Elp456.

C  Same as (A) using GST-tagged full-length Elp1 and truncations and deletions of Elp456.

D  Same as (A) using GST-tagged C-terminally truncated Elp1 in combination with truncated versions of Elp4.

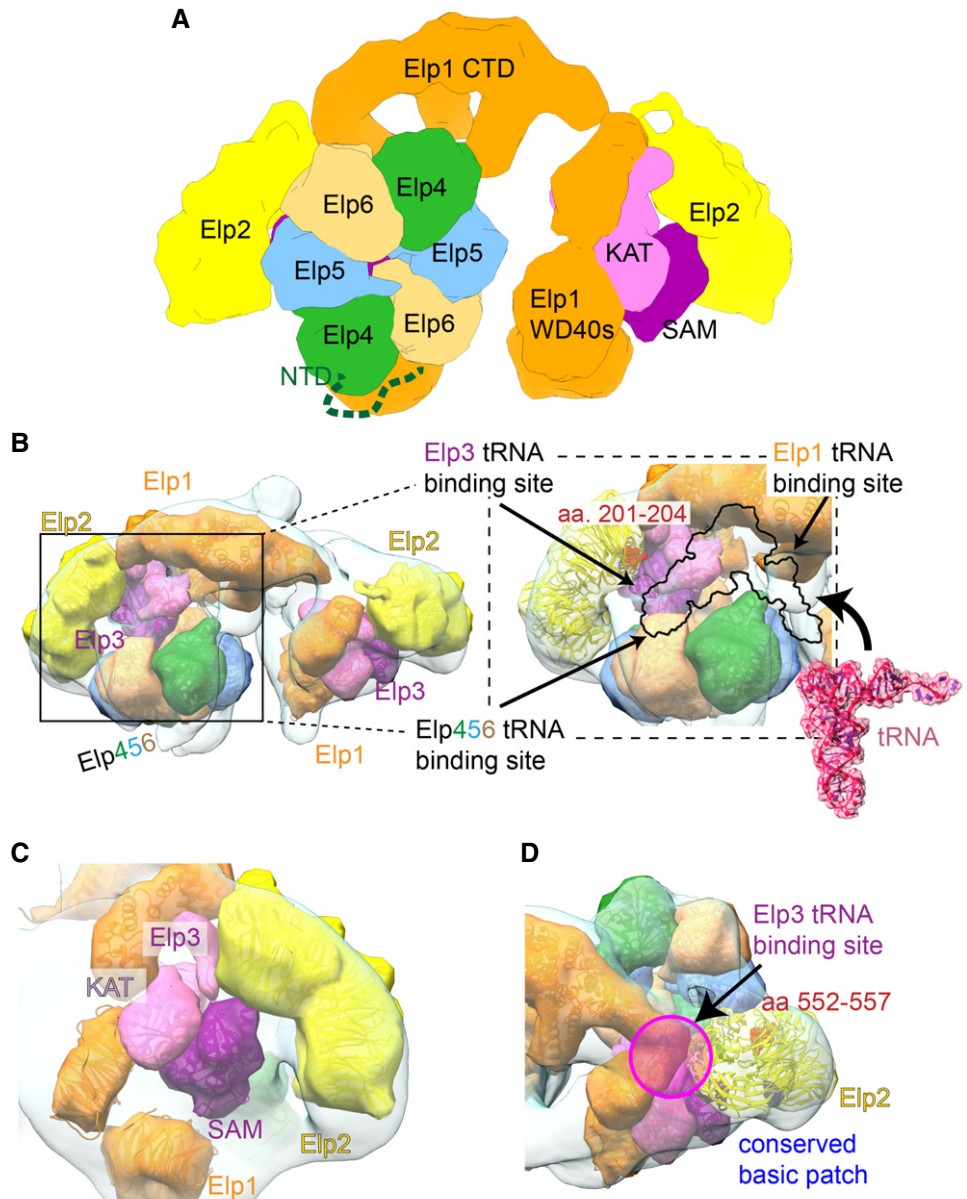

**Figure 6. Model of Elongator's tRNA modification reaction.**

A Overview of holoElongator and individual domains important for the interaction of Elp123 with Elp456. Elp1 (orange), Elp2 (yellow), Elp3 (pink), Elp4 (green), Elp5 (blue), and Elp6 (sand) are shown in surface representation to indicate the topological nature of our model. Individual domains are labeled.

B Same as (A) but tilted view to highlight the active site cavity (left). Close-up view of the active site in which all known tRNA binding sites and a region of Elp2 important for tRNA modification activity are highlighted. Individual structures of the subunits are shown in cartoon and surface representation.

C Close-up view on the localization of Elp3 and the orientation of its KAT and SAM domains in relation to Elp1 and Elp2.

D Close-up view on the relative spatial orientation of the conserved basic patch in Elp2 and the potential interaction site between Elp456 and a region of Elp2 important for tRNA modification activity. The potential tRNA binding site in Elp3 is indicated (arrow and circle).

strongly suggests a direct role of the Elp456 sub-complex in tRNA recruitment prior to the modification reaction or clearance of the tRNA after the reaction. In that respect, it is surprising that the Elp456 sub-complex interacts asymmetrically only with one side of the Elp123 sub-complex and it remains to be shown what triggers the polarized interaction with a symmetric Elp123 sub-complex. We currently hypothesize that binding of Elp456 to one side of Elp123 induces a conformational change within the holocomplex that

hinders a second copy of Elp456 to bind to the second lobe. This would be in accordance with the relative movement observed between the lobes when comparing the Elp123 and the holoElongator EM models, in which they come closer to each other and form a more compact structure. It is also worth mentioning that deletions of any of the six subunits lead to similar phenotypes [44], suggesting that despite the asymmetrical localization of Elp456, all components are equally important for Elongator activity.

In addition, residues 552–557 of Elp2, which were found to be important for tRNA modification, locate to the interfaces between Elp3 and Elp5 (Fig 6D). Although we show that mutating the residues of this patch (R626, R628, R654, R675) affects Elongator's activity, it is not involved in protein–protein interactions within the Elp123 sub-complex (Fig 6D). Therefore, this conserved basic patch of Elp2, which was also implicated in microtubule binding, could indeed be available for the interaction with other factors or stabilize the Elongator assembly in different conformational states.

Notably, we fitted a monomeric model of Elp3 using integrative modeling, although DmcElp3 was shown to form monomers and dimers in solution. Restraining yeast Elp3 to its equivalent dimeric conformation (as observed in the crystal structure of DmcElp3) led to several crosslink violations and to worse fits to the EM map. On the one hand, eukaryotic Elp3s might not need to directly dimerize, due to the presence and function of the other five subunits, partially absent in archaea and completely lacking in bacteria. On the other hand, other cofactors, for example, Kti11/Kti13, which are not present in our protein preparations, might promote dimerization of Elp3 or induce conformational changes of the whole complex. As we were neither able to reconstitute Elp2 containing complexes from bacterially expressed subunits nor found an interaction of Elp2 with any of the other five subunits in GST pull-down assays, it remains to be shown whether and which known [33,34] or unknown posttranslational modifications are responsible for the stable interaction of Elp2 with the other Elongator subunits.

The position of the active center in the model of the holocomplex provides important mechanistic insights on the tRNA modification activity. Previously, we showed that ATP binding promotes dissociation of tRNA from Elp456 [20]. Thus, we speculate that tRNA molecules are delivered to the Elp123 complex via initial binding to the Elp456 ring and that the intrinsic ATPase activity of Elp456 triggers the dissociation of tRNA from the ring, which allows the transfer from Elp456 to the Elp123 complex.

Last but not least, our structural analyses of Elp123 and holo-Elongator also confirm the finding that the C-terminus of Elp1 is essential for the assembly of the overall Elongator complex. Interestingly, this region is missing in a subpopulation of familial dysautonomia (FD) patients, due to a heterozygous mutation that leads to splicing defect and results in the expression of a substantially C-terminal truncation of Elp1 [45]. From our analyses, it becomes obvious that the presence of a truncated form of Elp1 would lead to a disassembly of the full complex and diminished tRNA modification activity [46].

# Materials and Methods

### Yeast strains and fermentation

*Saccharomyces cerevisiae* strains BS1173 (*MATa; ade2-1;his3-11,15; leu2-3,112; Δtrp1; ura3-1; can1-100; Elp1-TAP::TRP1*) and BSY2369 (*MATa; ade2-1;his3-11,15; leu2-3,112; Δtrp1; ura3-1; can1-100; Elp6-TAP::TRP1*) were generated as described previously [47,48]. Yeast cells were grown on a fresh YPD plate and then transferred to a 150-ml flask of YPD with 0.05% adenine sulfate and incubated for 24 h at 30°C and 180 rev/min. This pre-inoculum was seeded into 1 l of the same medium, which after 6–7 h incubation in identical conditions was used to inoculate 100 l YPDA. Cells were grown in a BIOSTAT D-DCU fermentor (Sartorius) for 16 h at 30°C and 150 rev/min to an $OD_{600}$ of 5–6, harvested by centrifugation and stored at −80°C until use.

### TAP purifications

Tandem-affinity purification purifications were performed as described previously [20]. In detail, 1,500 g (BSY2369 strain) and 390 g (BS1173 strain) of cells were suspended in buffer *A* (250 mM HEPES pH 7.9, 125 mM NaCl, 0.1% NP-40, 1 mM DTT, 10% glycerol, 1 mM PMSF) supplemented with protease inhibitor cocktail (Complete EDTA-free, Roche) and lysed at 4°C with glass beads in a BeadBeater (BioSpec). The soluble fraction obtained after centrifugation (1 h at 30,000 *g* in a Beckmann JA14 rotor) was incubated with 5 ml pre-equilibrated IgG Sepharose (GE Healthcare) for 4–6 h. After washing with buffer *B* (50 mM HEPES pH 7.9, 125 mM NaCl, 0.1% NP-40, 1 mM DTT, supplemented with protease inhibitor cocktail, and 10% glycerol for the BSY2369 strain) and buffer *C* (10 mM HEPES pH 7.9, 125 mM NaCl, 0.1% NP-40, 0.5 mM EDTA, 1 mM DTT, and 10% glycerol for the BSY2369 strain), the IgG beads were mixed with TEV protease and incubated overnight at 4°C. The supernatant was recovered and the resin was further washed with buffer *C*. Then, three column volumes of buffer *D* (10 mM HEPES pH 7.9, 125 mM NaCl, 1 mM MgAcetate, 1 mM imidazole, 2 mM $CaCl_2$, 10 mM 2-mercaptoethanol, and 10% glycerol for the BSY2369 strain) supplemented with 3 mM $CaCl_2$ (final concentration) were added to the sample that was subsequently incubated with 5 ml of Calmodulin Sepharose (GE Healthcare) for 2–4 h. After washing with buffer *D,* the sample was eluted in buffer *E* (10 mM HEPES pH 7.9, 125 mM NaCl, 1 mM MgAcetate, 1 mM imidazole, 2 mM EGTA, 10 mM 2-mercaptoethanol). The sample was concentrated to 0.3 mg/ml (Elp123) and 0.2 mg/ml (Elongator), crosslinked with 0.01% glutaraldehyde (Electron Microscopy Sciences) for 1 h at 4°C, and quenched with 40 mM Tris–HCl buffer (final concentration) for 10 min at 4°C. The final purification step comprises a gel filtration using a Superose 6 Increase 3.2/300 column (GE Healthcare) in buffer *F* (10 mM HEPES pH 7.9, 125 mM NaCl, 5 mM 2-mercaptoethanol). Crosslinked Elp123 complex was analyzed by Western blotting with TAP-tag antibody (Thermo Scientific) and anti-Elp2 antibody (polyclonal, 1:2,000).

### Individual proteins expression and purification

The codon-optimized sequences of Elp1, Elp2, Elp3, and Elp4 were subcloned into the pETM30 vector to obtain N-terminally 6xHis-GST-tagged proteins and transformed into BL21 pSarRare *E. coli*. Cultures were grown at 37°C until $OD_{600}$ ~1, and protein expression was induced using 0.5 mM isopropyl-β-D-thiogalactopyranoside (IPTG), incubated at 18°C overnight, and harvested the next morning by centrifugation. Pellets were resuspended in lysis buffer (50 mM Tris–Cl pH 7.5, 300 mM NaCl, 20 mM imidazole, 1 mM $MgCl_2$, 5% glycerol, 2 mM 2-mercaptoethanol, protease inhibitor tablets (Roche), DNase, and lysozyme) and lysed using a high-pressure homogenizer. The lysate was cleared by centrifugation at 15,000 *g* for 1 h at 4°C, and the supernatant was incubated with NiNTA resin for 2 h at 4°C. The bound protein was washed with wash buffer (50 mM HEPES pH 7.5, 300 mM NaCl, 1 mM $MgCl_2$,

5% glycerol, 2 mM 2-mercaptoethanol) and eluted with elution buffer (50 mM Tris pH 7.5, 300 mM NaCl, 250 mM imidazole, 1 mM 2-mercaptoethanol). Subsequently, the eluted protein was either dialyzed (20 mM Tris pH 7.5, 300 mM NaCl, 20 mM imidazole, 1 mM 2-mercaptoethanol) in the presence of TEV protease at 4°C overnight or directly applied to a S200 (26/60) gel filtration column (GE Healthcare) equilibrated in gel filtration buffer (20 mM HEPES pH 7.5, 150 mM NaCl, 5 mM DTT) to obtain GST-tagged versions. The cleaved tag was removed using a second NiNTA step and flow-through was also applied to gel filtration column. Fractions were analyzed by SDS–PAGE, pooled, and concentrated. Single amino acid substitutions were generated using the QuikChange mutation kit (Agilent Technologies). All mutant proteins were expressed and purified like the wild-type protein.

Full-length His6-tagged ScElp3 (ScElp3H6) was recombinantly expressed in *E. coli* together with GroEL (plasmid pGroEL kindly provided by Dr. Aguilar Netz, Marburg) and solubilized from inclusion bodies in buffer containing 8 M urea. Affinity purification on Ni-NTA-agarose was performed in the presence of 8 M urea followed by Fe/S cluster reconstitution *in vitro* in an anaerobic chamber [49]. Shortly, 53 mg of the apoprotein was first allowed to assemble iron at room temperature in a reaction volume of 150 ml containing 10 mM Tris–HCl, pH 7.5, 2 mM DTT, 8 M urea, and 220 µM ammonium Fe (III) citrate. After 5 min of incubation, S-adenosylmethionine (SAM) was added to a final concentration of 350 µM. 10 ml 10 mM Tris–HCl pH 7.5 supplemented with freshly prepared 20 mM Li2S and 2 mM DTT was added and folding was continued for 20 min. The resulting protein preparation was dialyzed against 10 mM Tris–HCl pH 7.5 and 150 mM NaCl and concentrated 40-fold by Amicon tubes with 50-kDa MWCO membrane before it was anaerobically applied to a gel filtration column (HiLoad™/60 Superdex 200 Prep. Grade) in 10 mM Tris–HCl, pH 7.5; 150 mM NaCl buffer supplemented with fresh 1 mM NaDTH to prevent oxidation of reconstituted Fe/S cluster. The reconstituted ScElp3H6 protein eluted as a single peak and was stored in anaerobic tubes.

## Crosslinking mass spectrometry analyses

50 µg (1 mg/ml) of purified Elongator complex was crosslinked by addition of an iso-stoichiometric mixture of H12/D12 isotope-coded, di-succinimidyl-suberate (DSS) or di-succinimidyl-glutarate (DSG, Creative Molecules). Equal amounts of crosslinker were added ten times every 4 min to a final concentration of 0.5–2 mM. The crosslinking reactions were allowed to proceed for 30 min at 37°C and quenched by addition of ammonium bicarbonate to a final concentration of 50 mM for 10 min at 37°C. Crosslinked proteins were denatured using urea and Rapigest (Waters) at a final concentration of 4 M and 0.05% (w/v), respectively. Samples were reduced using 10 mM DTT (30 min at 37°C), and cysteines were carbamidomethylated with 15 mM iodoacetamide (30 min in the dark). Protein digestion was performed first using 1:100 (w/w) LysC (Wako Chemicals GmbH, Neuss, Germany) for 3.5 h at 37°C and then finalized with 1:50 (w/w) trypsin (Promega GmbH, Mannheim, Germany) overnight at 37°C, after the urea concentration was diluted to 1.5 M. Samples were then acidified with 10% (v/v) TFA and desalted using MicroSpin columns (Harvard Apparatus). Cross-linked peptides were enriched using size exclusion chromatography

[50]. In brief, desalted peptides were reconstituted with SEC buffer (30% (v/v) ACN in 0.1% (v/v) TFA) and fractionated using a Superdex Peptide PC 3.2/30 column (GE) on a Ettan LC system (GE Healthcare) at a flow rate of 50 ml/min. Fractions eluting between 1 and 1.5 ml were evaporated to dryness and reconstituted in 50 µl 5% (v/v) ACN in 0.1% (v/v) FA.

Between 2 and 10% of the amount contained in the collected fractions were analyzed by liquid chromatography (LC)-coupled tandem mass spectrometry (MS/MS) using a nanoAcquity UPLC system (Waters) connected online to LTQ-Orbitrap Velos Pro instrument (Thermo). Peptides were separated on a BEH300 C18 (75 × 250 mm, 1.7 mm) nanoAcquity UPLC column (Waters) using a stepwise 60-min gradient between 3% and 85% (v/v) ACN in 0.1% (v/v) FA. Data acquisition was performed using a top-20 strategy where survey MS scans (m/z range 375–1,600) were acquired in the Orbitrap ($R = 30,000$) and up to 20 of the most abundant ions per full scan were fragmented by collision-induced dissociation (normalized collision energy = 40, activation $Q = 0.250$) and analyzed in the LTQ. In order to focus the acquisition on larger crosslinked peptides, charge states 1, 2 and unknown were rejected. Dynamic exclusion was enabled with repeat count = 1, exclusion duration = 60 s, list size = 500, and mass window ± 15 ppm. Ion target values were 1,000,000 (or 500 ms maximum fill time) for full scans and 10,000 (or 50 ms maximum fill time) for MS/MS scans. The sample was analyzed in technical duplicates (for DSG crosslinker) or triplicates (for DSS crosslinker). To assign the fragment ion spectra, raw files were converted to centroid mzXML using the Mass Matrix file converter tool and then searched using xQuest [51] against a fasta database containing the sequences of the crosslinked proteins. Posterior probabilities were calculated using xProphet, and results were filtered using the following parameters: FDR = 0.05, min delta score = 0.95, MS1 tolerance window of 4–7 ppm, ld score > 30. The mass spectrometry proteomics data have been deposited to the ProteomeXchange Consortium via the PRIDE [52] partner repository with the data set identifier PXD005251.

## *In vitro* interaction assays

Typically, 20 µg of GST-tagged proteins or mutants and increasing amounts of untagged binding partner protein was incubated at 4°C for 2 h with glutathione Sepharose in 20 mM Tris pH 7.5, 150 mM NaCl, 5 mM DTT, and 0.1% Tween-20. The beads were washed five times with incubation buffer and subsequently resuspended in SDS loading buffer. Inputs and bound proteins were separated using denaturing SDS–PAGE and visualized using Coomassie blue stain.

## ITC measurements

Isothermal titration calorimetry was performed with a VP-ITC Microcal calorimeter (Microcal, Northampton, MA, USA). To measure the Elp1 Elp4 interactions, Elp1 samples were dialyzed extensively against ITC buffer (20 mM Tris–HCl, pH 7.5, 150 mM NaCl, 2 mM β-mercaptoethanol). Lyophilized synthesized peptides (Peptide Specialty Laboratories, Germany) were solubilized in ITC buffer just before use. Protein/peptide concentration in the cell was 10 µM and 100 µM in the injection syringe. The data were analyzed using Origin software (GE Healthcare).

## Elp123 negative-stain EM and image processing

3.5 µl aliquots of freshly purified Elp123 complex were applied to glow discharged carbon copper-collodion (Sigma) grids for 2 min and stained with a 1% uranyl acetate solution (w/v). A total of 216 images were collected on a FEI Tecnai Spirit microscope operated at 120 kV using a Gatan Ultrascan 4000 camera at a final magnification of 49,000×. The defocus of the selected images ranged from 1.5 to 3.5 µm and the pixel size corresponded to 2.2 Å/pixel. Contrast transfer function parameters were estimated using CTFFIND3 [53].

A total of 50,034 particles were semi-automatically picked using EMAN2 [54] and subjected to reference-free classification inside RELION [55], from which 45,650 particles were selected for subsequent processing. As a starting model for the 3D classification, we used first geometric shapes of a sphere and a cylinder slab. Second, 120 tilt pairs (0 and 55° tilt) were collected on a FEI Titan Krios at 1 µm underfocus at a pixel size of 3.78 Å/pix. The picked 6,849 particle pairs were subjected to the random conical tilt procedure as implemented in XMIPP [56]. Refining the geometric shapes as well as an initial RCT model resulted in the same characteristic two-lobed structure. Subsequently, several rounds of 3D classification were performed and the best classes were selected based on the comparison between the back-projections and the class averages, as well as on their symmetric features. The first round of 3D classification yielded two good classes (18,588 particles), and after a second round, 6,469 particles were selected. In the third round of 3D classification, the best class showed an apparent twofold symmetry axis (though no symmetry was previously imposed) and was independently refined with and without C2 symmetry, yielding two equivalent Elp123 reconstructions. The "gold standard" refinement with C2 symmetry applied yielded a final reconstruction (2,051 particles) at 27 Å resolution, based on the FSC = 0.143 criterion and 35 Å according to the FSC = 0.5 threshold.

In addition, a fourth round of 3D classification yielded a class lacking one of the lateral densities and thus showing no symmetry, which would correspond to the partial Elp123 sub-complex. The refinement step yielded a final reconstruction (1,946 particles) at 31 Å resolution based on the FSC = 0.143 criterion and 35 Å according to the FSC = 0.5 threshold.

The back-projections of the EM models were obtained using the "create projection library" tool in XMIPP [56].

## HoloElongator negative-stain EM and image processing

3.5 µl aliquots of freshly purified holocomplex were applied to glow discharged carbon copper-collodion (Sigma) grids for 2 min and stained with a 1% uranyl acetate solution (w/v). A total of 204 images were collected on a FEI Tecnai Spirit microscope operated at 120 kV by using a Gatan Ultrascan 4000 camera at a final magnification of 49,000× that corresponds to a pixel size of 2.2 Å/pixel. The defocus of the selected images ranged from 1.5 to 3.5 µm and the pixel size corresponded to 2.2 Å/pixel. Contrast transfer function parameters were estimated using CTFFIND3.

A total of 22,876 particles were semi-automatically picked using EMAN2 and subjected to reference-free classification inside RELION, from which 22,064 particles were selected. As a starting model for the 3D classification, we used the Elp123 model, low-pass-filtered to 60 Å. Four rounds of 3D classification were performed and the best

classes were selected based on the comparison between the 2D averages and the model back-projections. After the first round of 3D classification, 7,234 particles were subjected to a second round in which the major class gathered 5,057 particles. In the third round of 3D classification, 3,787 particles were selected from one class. After the fourth round of 3D classification, the "gold standard" refinement yielded a final reconstruction (3,164 particles) at 31 Å resolution based on the FSC = 0.143 criterion and 36 Å according to the FSC = 0.5 threshold. A tilt-pair test [54,57,58] was performed to confirm the handedness and correctness of our holoElongator model.

## Structural modeling

As initial structures for the modeling, we used the available crystal structures of Elongator subunits or homology models if crystal structures were not available. For WD40 domains of Elp1, we built homology models based on coatomer β'-subunit (PDB code: 3MKQ, chain A) based on the alignment generated by HHpred server [59]. Due to low sequence similarity between Elp1 and coatomer β'-subunit, these models should be regarded as low quality with the confident assignment of the fold but uncertain sequence register. For Elp1-CTD, we used the yeast crystal structure of Elp1-CTD (PDB code: 5CQS) with the missing region comprising residues 739–919 added from human Elp1-CTD structure (PDB code: 5CQR) by homology modeling. For Elp2, we used the structure published in this work. For Elp3, we built a homology model built based on Elp3 from *Dehalococcoides mccartyi* (PDB code: 5L7J). This model is predicted to be of high confidence owing to the high sequence conservation between yeast and *D. mccartyi* Elp3. For Elp4, Elp5, and Elp6, we used the crystal structure of Elp456 hexamer (PDB code: 4A8J). Short loops and regions missing in the structures were added to the models as flexible loops and helices based on secondary structure predictions if they formed crosslinks potentially useful for model. These included, for example, residues 74–86 and 375–406 of Elp3 and 1,244–1,252 of Elp1. The homology modeling was performed using MODexplorer [60], HHpred server [59], and Modeller [61]. The secondary structure predictions were generated using GeneSilico MetaServer [62].

Fitting of each Elongator domain separately to the EM maps and calculations of *P*-values for the fitting scores was performed as described previously [63,64] using UCSF Chimera software [65]. Briefly, each domain was fitted using global search with an arbitrarily large number of 10,000 random initial placements and a normalized cross-correlation score as a fitting metric. The fits were then clustered, in the case of Elp123 sub-complex taking the twofold symmetry axis into account. The *P*-values for the cross-correlation scores were calculated by transforming to *z*-scores (Fisher's *z*-transform) and centering, and fitting an empirical null distribution from which two-sided *P*-values were computed [64].

Automated integrative modeling was performed using our previously published workflow [39] based on the UCSF Chimera, Integrative Modeling Platform (IMP) package [40], version g72059d2, and Python Modeling Interface (PMI) library (https://github.com/salilab/pmi), version gfe8bea8.

For modeling the Elp123 sub-complex, each domain of Elp1, Elp2, and Elp3 was firstly independently fitted to the EM map using UCSF Chimera. The fitting was performed using a global search with an

100,000 random initial placement leading to, after clustering, 10,000–30,000 alternative fits for each domain. To ensure broad coverage of fitting orientations and overcome that negative-stain maps often contain artificially dense regions in the center of the densities, which tend to "attract" the fits, the fitting was performed with shorted optimization of 100 steps and clustering with very low angular and the translational thresholds (cluster angle 1°, cluster shift 1 Å). For subsequent steps, we took the top scoring fit of Elp1-CTD, since the fit was unambiguous for this domain, and 10,000 top scoring fits for other domains. The fit Elp1-CTD was additionally optimized with crude flexible fitting by generating a series of conformations Normal Mode Analysis and selecting the conformation optimally fitting the EM map (using the ElNemo [66] server). Then, we generated 10,000 configurations of all Elp1, Elp2, and Elp3 domains by recombining the above fits using simulated annealing Monte Carlo optimization. Each configuration was generated by an independent Monte Carlo optimization comprising 60,000 steps. The scoring function for the optimization was a linear combination of the normalized EM cross-correlation scores of the precalculated domain fits, crosslinking restraint, domain connectivity restraint, and clash score (see [39] for the implementation details). All domains were treated as rigid bodies, including the N-terminal helices of Elp3, which consisted rigid bodies moving independently on each other, except short loops and linkers connecting the structured regions. Loops and linkers that were missing in the starting structures were excluded. The structures were simultaneously represented at two resolutions: in Cα-only representation and a coarse-grained representation, in which each 10-residue stretch was converted to a bead. The 10-residue bead representation was used for the clash score to increase computational efficiency; the Cα-only representation was used for crosslinking and domain connectivity restraints. Since the EM restraint was derived from the original EM fits generated with UCSF Chimera, it was derived from the full atom representation.

From the 10,000 configurations, the top scoring model was selected and refined in Cα representation. The refinement was performed using the same scoring function as above, but instead of generating configuration from the predefined sets of fits, the domains were allowed to move in continuous space and the EM cross-correlation score was recomputed during the optimization. The resulting Cα model was converted to the full atom representation with Modeller by using the starting domain structures and the Cα model as constraints.

For modeling the holoElongator, we used the top scoring Elp123 model and the Elp456 crystal structure as the starting structures. Firstly, the Elp123 model was fitted to the holoElongator map as a rigid body using UCSF Chimera. From the fit, it was apparent that the two lobes of Elp123 change relative orientation in the holoElongator. Thus, based on the fit, we identified a hinge region around aa. 945–960 of Elp1, and divided the Elp123 model into two lobes, and fitted the lobes independently as rigid bodies. Then, the Elp456 was fitted to the map and the entire model was refined using the same refinement procedure as for refining the Elp123 model. Mapping of the sequence conservation onto protein structures was performed using ConSurf server [67] based on the multiple sequence alignments of Elp families retrieved from eggNOG database [68]. The electrostatic potential was mapped to protein structures using Pymol and APBS [69].

## Crystallization and structure determination of Elp2

Native and selenomethionine substituted crystals were grown at 18°C using the hanging drop vapor diffusion method. Purified Elp2 protein in gel filtration buffer was concentrated to 10 mg/ml and combined with equal volume of 100 mM Tris pH 7.5, 10% PEG 6K, 1,750 mM NaCl. Crystals grew until day 2, were cryo-protected with 25% glycerol, and subsequently flash-frozen in liquid nitrogen. Native and selenomethionine data sets for Elp2 were collected at ESRF beamline ID14-4 equipped with a CCD camera. Data processing was performed using XDS [70]. SAD phases were calculated from identified selenium sites using autoSHARP [71]. The resulting electron density map after solvent flattening was of good quality and an initial model was built and subsequently refined using Phenix [72]. The geometry of the models was validated using Molprobity [73]. Model figures and superimpositions were prepared using Pymol (www.pymol.org) and Coot [74].

## Thermofluor analyses of Elp2 mutants

Thermostability of purified Elp2 proteins was analyzed using thermofluor technology [75]. In detail, 20–50 μM of purified protein was incubated with SYPRO Orange in 20 mM HEPES pH 7.5, 150 mM NaCl, and 5 mM DTT. Using a StepOne Plus Real-Time PCR system (Applied Biosystems), the samples were subjected to a temperature gradient from 20°C to 95°C (over the period of 2 h) while simultaneously measuring the emitted fluorescence signal ($\lambda$ = 590 nm).

## Phenotypic analyses Elp2 mutants

A complementing clone encoding wild-type ELP2, pRS315-Elp2 was obtained from A. Byström. A derivative, pBS4396, containing a C-terminal TAP-tag [35] was constructed. Most of the mutant derivatives were obtained by inserting in the cognate vectors three PCR fragments covering roughly the promoter region, coding sequence, and TAP-tag/terminator region using the Cold Fusion strategy (System BioSciences). For point mutants, the resulting plasmids were generated: pBS4517 (Elp2 R626A + D627A + R628A), pBS4519 (Elp2 Y440A + D441A), pBS4520 (Elp2 D458A + E459A + K460A), pBS4521 (Elp2 R675A), pBS4522 (Elp2 R654A), pBS4523 (Elp2 H201A + E202A + D203A + W204A), pBS4592 (Elp2 E552A-K553A-L554A-Y555A-G556A-H557A), pBS4593 (Elp2 E597A-I598A-K599A), pBS4594 (Elp2 R322A-E325A), pBS4857 (Elp2 R544A + H545A).

## Accession numbers

The atomic coordinates and structure factors for Elp2 (PDB ID 5M2N) have been deposited with the European Protein Data Bank (PDBe). The EM density maps of Elp123 (EMDB EMD-4151), holoElongator (EMD-4152), and partial Elp123 (EMD-4153) have been deposited with the EMData Bank (EMDB).

**Expanded View** for this article is available online.

## Acknowledgements

We acknowledge support by the EMBL Heidelberg Crystallization Platform, the Protein Expression and Purification Core facility, and the Proteomics

Core Facility and technical support by V. Rybin, R. Wetzel, and H. Groetsch. We would like to thank W. Hagen and H. Khatter for excellent technical support on EM data collection and M. Vorländer for providing purified RNA polymerase III. We acknowledge B. Webb, Ch. Greenberg, and A. Sali for support regarding the Integrative Modeling Platform and A. Byström for providing constructs for complementation studies. We also acknowledge access and support by the EMBL/ESRF Joint Structural Biology Group at ESRF beamlines. This work was also supported by the Ligue contre le Cancer (Equipe labellisée 2014) (BS), the Centre National pour la Recherche Scientifique (BS), the CERBM-IGBMC, the project Elongator from the Agence Nationale pour la Recherche (grant ANR-13-BSV8-0005-01) (B.S.), the grant ANR-10-LABX-0030-INRT managed under the program Investissements d'Avenir ANR-10-IDEX-0002-02 (BS), the EMBL Interdisciplinary Postdoc Programme under Marie Curie COFUND actions and EMBO Short Term Fellowship (JK), the OPUS10 grant UMO-2015/19/B/NZ1/00343 from the National Science Centre (SG), SFB 648 (KDB) and the grant "Regulation of Elongator and DPH complexes by the Kti11/Kti13 heterodimer" (BR921/9-1 and Mu3173/2-1) from the Deutsche Forschungsgemeinschaft (KDB, MID, CWM).

## Author contributions

MID and SG established protein purification procedures; MID performed the EM characterization and analyses. JK conducted all molecular modeling and fitting analyses; OK-R, CF, and BS created TAP and mutant strains and performed initial complex purifications as well as phenotypical characterization; SG performed all biochemical and biophysical characterization and solved the Elp2 structure. SG and AD initially characterized protein samples by electron microscopy with the help of NAH and CS; AO performed XLMS experiments, mass spectrometry, and data analyses with the support of SG and MB; OFO and KDB purified and reconstituted bacterially expressed ScElp3; MID, JK, SG, BS, and CWM designed experiments and analyzed the data; MID, JK, SG, and CWM wrote the manuscript with input from the other authors.

## Conflict of interest

The authors declare that they have no conflict of interest.

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
