## [Review Process File · EMBO Reports]

Manuscript EMBO-2016-43353

Architecture of the yeast Elongator complex

Maria I. Dauden, Jan Kosinski, Olga Kolaj-Robin, Ambroise Desfosses, Alessandro Orii, Celine Faux, Niklas A. Hoffmann, Osita F. Onuma, Karin D. Breunig, Martin Beck, Carsten Sachse, Bertrand Seraphin, Sebastian Glatt, Christoph W. Müller

Corresponding author: Christoph Müller, European Molecular Biology Laboratory

Review timeline:	Submission date:	16 September 2016
	Editorial Decision:	07 October 2016
	Revision received:	20 October 2016
	Editorial Decision:	08 November 2016
	Revision received:	08 November 2016
	Accepted:	14 November 2016

Editor: Esther Schnapp

Transaction Report:

1st Editorial Decision

07 October 2016

Thank you for the transfer of your research manuscript to EMBO reports. We have now received the comments from the referees that were asked to assess it, and that are pasted below.

As you will see, both referees acknowledge that the findings are interesting and novel. They have a few suggestions for how the study could be further strengthened and improved, and I think that all of them should be addressed, except point 3 of referee 1, which is outside the scope of this study.

Given these constructive comments, we would like to invite you to revise your manuscript with the understanding that the referee concerns must be fully addressed and their suggestions taken on board. Please address all referee concerns in a complete point-by-point response. Acceptance of the manuscript will depend on a positive outcome of a second round of review. It is EMBO reports policy to allow a single round of revision only and acceptance or rejection of the manuscript will therefore depend on the completeness of your responses included in the next, final version of the manuscript.

Given the competitive situation, I suggest that you resubmit your manuscript in 2 weeks from now, so please contact me latest on the 23rd of October. You can either publish the study as a short report or as a full article. For short reports, the revised manuscript should not exceed 25,000 characters (including spaces but excluding materials & methods and references) and 5 main plus 5 expanded view figures. The results and discussion sections must further be combined, which will help to shorten the manuscript text by eliminating some redundancy that is inevitable when discussing the

same experiments twice. For a normal article there are no length limitations, but it should have more than 5 main figures and the results and discussion sections must be separate. In both cases, the entire materials and methods must be included in the main manuscript file. Supplementary material is called Expanded View now. Please upload EV figures and tables as individual files and add the EV figure legends at the end of the main manuscript file.

Please note that our papers have a numbered reference style, which can be found in EndNote.

As I said, we have another manuscript in the pipeline reporting the yeast Elongator structure that will be accepted soon. I am attaching the pdf file of the other manuscript to this email; let me know if you would like to see the expanded view figures too. Only minor changes to the format will be required. Please discuss the related study in your manuscript and include a proper citation. Yip et al will do the same with your paper. We will proceed with the online publication of the other study, but will try to include both papers in the same issue of the journal, which will be January 2017 or later.

REFEREE REPORTS

Referee 1:

Dauden and colleagues here describe the architecture of the elongator complex, using a combination of negative stain electron microscopy, crosslinking mass spectrometry, and biochemical assays. Despite its important cellular function in tRNA modification, currently only crystal structures of parts of Elongator are known: the crystal structure of the hexameric Elp456 complex (Glatt and Müller, NSMB 19: 314-320 (2012); Lin et al., JBC 287: 21501-21508 (2012)), Elp2 (Dong et al., Structure 23, 1078-1086 (2015) and a dimer of the C-terminal domain of Elp1 (Xu et al., PNAS 112: 10697-10702 (2015)). The authors present a low resolution EM structure of Elp1-6 ('holo-Elongator') and Elp123 and together with crosslinking data propose an integrative model of its architecture. The major findings are: (i) the symmetric subunit organization of Elp123 and asymmetric localisation of the Elp456 hexamer, (ii) a thorough analysis of subunit interaction within Elongator, and (iii) identification of the binding site for tRNA substrates in the resultant Elongator model. The manuscript is well written and the data clearly presented. Subject to satisfactory revision, this manuscript will make an important contribution to our understanding of Elongator structure and function in the cell. A revised version should aim to experimentally verify at least the Elp1 subunit location and to determine the handedness of the EM reconstruction using tilt-pairs. Ideally, the authors should explore the importance of Elongator asymmetry for its function in vitro.

Major points:

1. The authors apply an integrative structural biology approach to determine the subunit architecture of the yeast Elongator complex. Confidence in the model would be greatly strengthened by further EM validation. The authors could use negative stain EM to confirm the location of at least the dimerization hinge of Elp1, for example by localizing its C-terminus with an antibody to the existing TAP-tag, and ideally also by localizing its N-terminus. The authors could further verify that Elp2 is indeed in the periphery of the complex, using an appropriate immune-labelling or subunit-tagging strategy.
2. The computational efforts to identify the correct handedness of the reconstructions are suggestive, but not sufficiently convincing at the low resolution reported (27 \approx). The authors should carry out a tilt-pair validation to determine the correct handedness with absolute certainty.
3. The authors propose that Elp456 delivers the tRNA substrate, which is then modified by Elp3. Is the purified holo-Elongator therefore a post-catalysis state after tRNA release, to which Elp456 stably binds? How is Elp456 dissociated to enable the next Elp456 subcomplex to associate? For example, if the nucleotide state of the associating Elp456 is different (ADP?) from the bound, post-transfer, Elp456 (ATP?), can this displace the Elp456 molecule on the same Elp123 wing or would it bind the other Elp123 wing, causing the allosteric release of the other Elp456 molecule? What is the nucleotide state of the Elp345 hexamer in holo Elongator? These and similar questions could be

addressed by the authors in vitro using purified proteins. For example, the authors could mutate the Elp1 dimerization interface to obtain monomeric Elp123 bound to the Elp456 hexamer and compare the activity of this complex (ATPase and tRNA modification) to that of the full Elongator.

4. It would be extremely helpful to see the charge distribution and sequence conservation of the proposed tRNA binding cavity, and also sequence conservation at intersubunit interfaces within holo Elongator. This may further inform on the accuracy of the architectural model.

Minor points:

1. The authors should provide 2D Euler plots for the angular distribution of refined Elp123, and holo-Elongator reconstructions.
2. To better judge the quality of the protein crosslinking data the authors should include the following information in supplementary table 2: linked residues, sequence of both peptides, m/z, charge, and ppm error of the identified precursor ion.
3. Did the authors observe crosslinks that confirm the Elp1 C-terminal dimerization interface observed in the crystal structure by Xu et al. PNAS?
4. Does the holo Elongator complex co-purify with any tRNA species? See also Major point 3.

Referee 2:

In their manuscript Dauden et al. report the first topological model of the full Elongator complex and the Elongator123 complex required for xm5 formation on wobble uridine of tRNA. The authors generated the model by negative stain EM on TAP-purified complexes from bakers yeast. The EM scaffold was estimated as 31 \approx and 27 \approx . To refine placements into the Holo structure, they use cross-linking mass spectrometry to determine relative subunit orientation in the structure. This allowed for the fit of recently published and newly generated structures leading to a full model. In addition to the modeling work the authors tested some of the interactions that they observed in the MS and predicted from the model by GST-pulldown. The new model also fits a tRNA that was modeled into a site previously described as critical for tubulin-acetylation. According to an indirect assay they find suggestive evidence that the mutations in the domain impair tRNA modification.

The holoElongator complex consists of 2x each subunit of Elp1-6. Older hypothetical models (also by the authors) had placed the ring of Elp456 in the middle of a putative holo-structure surrounded by two copies of Elp123. The current manuscript disproves the idea and shows that the holo-complex is asymmetrical and that a wing-shaped Elongator123 forms the structural basis of the holo-complex, where an Elp456 ring binds to one of the Elp123 subunits. Finally, the model may provide an explanation why the mutation prevalent in most patients of familial dysautonomia leads to a dysfunctional Elongator complex.

This is a surprising finding that has implications on how the Elongator complex acts as a molecular machine. There are conflicting reports about different functions of the Elongator complex. Careful analysis of the structural determinants of the complex will ultimately solve these discrepancies. However, the authors are careful not to overstate their finding. The work is suited for a publication in EMBO Reports if some points are addressed.

- Can the authors discuss why there are so few elp4, 5,6 crosslinks?
- The authors claim that "Elp1 interacts with KAT domain of Elp3, but shows only very weak interaction with the SAM domain of Elp3" (Figure 4A). This seems an overstatement as a band of Elp1 is seen in both pulldowns. Please Change.
- The authors claim that mutant Elongator leads to phenotypes typical for RNA modification defects. Please state clearly that this is not a proof of a modification defect, but rather an indirect assessment of tRNA modification. The main weakness is that the authors do not provide direct evidence for the absence of xm5 formation. But this may be beyond the scope of the article.

A general comment: The preparations of protein complexes do not appear completely clean or not well labeled in Figure 1A. However, the sample on the grid appears homogeneous. What are the additional bands and why don't they matter.

1st Revision - authors' response

20 October 2016

Referee 1:

Dauden and colleagues here describe the architecture of the elongator complex, using a combination of negative stain electron microscopy, crosslinking mass spectrometry, and biochemical assays. Despite its important cellular function in tRNA modification, currently only crystal structures of parts of Elongator are known: the crystal structure of the hexameric Elp456 complex (Glatt and Müller, NSMB 19: 314-320 (2012); Lin et al., JBC 287: 21501-21508 (2012)), Elp2 (Dong et al., Structure 23, 1078-1086 (2015) and a dimer of the C-terminal domain of Elp1 (Xu et al., PNAS 112: 10697-10702 (2015)). The authors present a low resolution EM structure of Elp1-6 ('holo-Elongator') and Elp123 and together with crosslinking data propose an integrative model of its architecture. The major findings are: (i) the symmetric subunit organization of Elp123 and asymmetric localisation of the Elp456 hexamer, (ii) a thorough analysis of subunit interaction within Elongator, and (iii) identification of the binding site for tRNA substrates in the resultant Elongator model. The manuscript is well written and the data clearly presented. Subject to satisfactory revision, this manuscript will make an important contribution to our understanding of Elongator structure and function in the cell. A revised version should aim to experimentally verify at least the Elp1 subunit location and to determine the handedness of the EM reconstruction using tilt-pairs. Ideally, the authors should explore the importance of Elongator asymmetry for its function in vitro.

Major points:

1. The authors apply an integrative structural biology approach to determine the subunit architecture of the yeast Elongator complex. Confidence in the model would be greatly strengthened by further EM validation. The authors could use negative stain EM to confirm the location of at least the dimerization hinge of Elp1, for example by localizing its C-terminus with an antibody to the existing TAP-tag, and ideally also by localizing its N-terminus. The authors could further verify that Elp2 is indeed in the periphery of the complex, using an appropriate immune-labelling or subunit-tagging strategy.

Response: We have previously attempted to localize Elp1, Elp2, Elp3 and Elp5 in the holoElongator density using purified GFP-tagged versions of these subunits in similar negative stain EM approaches. Unfortunately, only the data for the Elp2-GFP fusion strain were of sufficient quality to assess the localization of Elp2 in the Elp123 sub-complex. Using 2D classification and 3D reconstruction allowed us locating the GFP-labeled Elp2 subunit at the periphery of the complex. The position of the Elp2 subunit is consistent with the position suggested by a reconstruction of Elp123 lacking one of the two Elp2 subunits (Fig. EV4, B-E). Compared to this Elp123 reconstruction, the GFP-labeling results are of lower quality and we prefer not to include them into the main manuscript, but nevertheless want to present them to the referees (see Elp2-GFP Figure below). For the other GFP-fusion strains we could not identify additional densities clearly enough to unambiguously locate the respective subunits. The analyses of these datasets and the detection of relatively small additional GFP densities is hindered by the obtained resolution, the observed relative movement of the two Elp123 lobes and other structural heterogeneity/flexibility in various regions of the complex.

Figure Elp2-GFP: Elp2GFP negative stain and image processing. A. Comparison between reference-free class averages of Elp2-GFP sample and wild type (wt). The back projections of the corresponding 3D models are shown in the lower row. The additional densities present in the Elp2GFP are labelled. Scale bar, 20 nm. B. Comparison between EM reconstructions of the Elp2GFP-Elp123 sub-complex and the Elp123 sub-complex, filtered at 38 Å resolution. The extra length with respect to the wt is highlighted in orange.

2. The computational efforts to identify the correct handedness of the reconstructions are suggestive, but not sufficiently convincing at the low resolution reported (27Å). The authors should carry out a tilt-pair validation to determine the correct handedness with absolute certainty.

Response: We agree with the referee that determining the corrected handedness of the structure is a very important point in a low resolution EM reconstruction. Following the suggestion of the referee we have carried out a tilt pair validation approach that clearly confirms the correct handedness of our reconstruction (Fig. EV2 E) with RNA Polymerase III as a reference sample of known handedness. In addition, we have further improved the graphical representations in Fig. EV2 (panels C and D) by adding orthogonal views to better demonstrate that the crystal structure of the Elp1-CTD fits much better into the correct "hand" of the EM volumes compared to its mirrored reconstruction. In summary, we have experimentally confirmed the correct handedness and present this data in the newly introduced panel E in Fig. EV2. We have added the following sentence on page 6 - "In addition, we used an experimental tilt pair validation approach to confirm the handedness and correctness of our 3D reconstruction (Fig. EV2E)."

3. The authors propose that Elp456 delivers the tRNA substrate, which is then modified by Elp3. Is the purified holo-Elongator therefore a post-catalysis state after tRNA release, to which Elp456 stably binds? How is Elp456 dissociated to enable the next Elp456 subcomplex to associate? For example, if the nucleotide state of the associating Elp456 is different (ADP?) from the bound, post-transfer, Elp456 (ATP?), can this displace the Elp456 molecule on the same Elp123 wing or would it bind the other Elp123 wing, causing the allosteric release of the other Elp456 molecule? What is the nucleotide state of the Elp345 hexamer in holo Elongator? These and similar questions could be addressed by the authors in vitro using purified proteins. For example, the authors could mutate the Elp1 dimerization interface to obtain monomeric Elp123 bound to the Elp456 hexamer and compare the activity of this complex (ATPase and tRNA modification) to that of the full Elongator.

Response: Referee 1 suggests characterizing different functional states of Elongator by mutating the Elp1 dimerization interface and to study monomeric Elp123 bound to the Elp456 hexamer. These are excellent suggestions - however, as described in the manuscript we cannot reconstitute Elongator using recombinant proteins, but instead use endogenous TAP-tagged Elongator that we purify directly from large quantities of yeast (~500 g - 1kg yeast paste). In addition, no tRNA activity assay of an in vitro reconstituted Elongator complex has been reported so far. Given the experimental difficulties obtaining pure wild type and mutant Elongator and considering the unclear outcome of these experiments, we believe that the suggested experiments correspond to a full research project on its own and go far beyond the scope of the current manuscript.

4. It would be extremely helpful to see the charge distribution and sequence conservation of the proposed tRNA binding cavity, and also sequence conservation at intersubunit interfaces within holo Elongator. This may further inform on the accuracy of the architectural model.

Response: Following the request of the referee, we now show charge distribution and sequence conservation of the proposed tRNA binding cavity in the newly introduced Appendix Fig. S3. In addition, we also include sequence conservation at inter-subunit interfaces in the same Appendix Figure S3. We refer to this new figure in two sections of the text on page 14/15 - "In detail, the active site of Elp3 is located in the conserved cavity formed by Elp3, the N-terminal WD40 domain of Elp2, and the α -solenoid domain of Elp1-CTD (Appendix Fig. S3A,B)." and "Overall, our model agrees with the independently identified protein-protein interfaces between Elp1, Elp3 and Elp456 and with the relative orientation of the conserved regions of all individual subunits (Appendix Fig. S3C,D,E)."

Minor points:

1. The authors should provide 2D Euler plots for the angular distribution of refined Elp123, and holo-Elongator reconstructions.

Response: 2D Euler plots for the angular distribution of refined holo-Elongator and Elp123 reconstructions have been included as new panels Fig. EV2B and Fig. EV3D, respectively. In addition, we refer to these new figure panels in two sections of the text on page 6: "The Elongator model shows an estimated resolution of 31 Å based on the Fourier Shell Correlation (FSC; Fig. EV2A) and although the angular assignment of particle orientations is well distributed, we observed a preferential orientation, a commonly observed phenomenon in negative stain EM (Fig. EV2B)." and page 7 "The back-projections of the 3D model correlate well with the class averages (Fig. 2C) and the overall angular assignment of particle orientations is well distributed (Fig. EV3D)."

2. To better judge the quality of the protein crosslinking data the authors should include the following information in supplementary table 2: linked residues, sequence of both peptides, m/z, charge, and ppm error of the identified precursor ion.

Response: All additional details of the identified cross-linked peptides (e.g. linked residues, sequence of both peptides, m/z, charge, and ppm error of the identified precursor ion) are now available in an additional source file. Appendix Table 2 is unchanged as it contains the most relevant information. We have added a note, which refers to the provided source file: "Please see source data file for a complete list and additional specifications of the detected cross links."

3. Did the authors observe crosslinks that confirm the Elp1 C-terminal dimerization interface observed in the crystal structure by Xu et al. PNAS?

Response: We do not identify any crosslinks that would directly confirm the dimerization. Even though we observe some Elp1-Elp1 crosslinks at the lower crosslink confidence thresholds, all of them can be explained by crosslinks within the same molecule. However, we would like to note that Xu et al. PNAS crystallized Elp1 dimer from two different species (yeast and human), and in both structures the dimeric interface is very similar. In addition, the dimeric structure fits very well to our EM map (see Fig. EV2C). Thus, we believe that the high-resolution crystal structure of the Elp1 dimerization domain in its CTD accurately describes its conformation within the Elp123 sub-complex and holoElongator.

4. Does the holo Elongator complex co-purify with any tRNA species? See also Major point 3.

Response: Using silver stain analyses of gel separated TAP purified fractions we could not detect any unassigned band that could correspond to tRNA. In addition, the Elp6TAP sample (holoElongator) shows a 260/280 nm UV Abs ratio before GF of around 0.7 suggesting that there are some substoichiometric nucleic acid contaminations. For the Elp1TAP purification the 260/280 ratio after crosslinking and GF was also ~0.7 (as for the non-cross-linked sample). Therefore, we have no strong evidence that Elongator co-purifies with tRNAs, but cannot exclude that a certain sub-fraction of particles has tRNA bound.

Referee 2:

In their manuscript, Dauden et al. report the first topological model of the full Elongator complex and the Elongator123 complex required for xm5 formation on wobble uridine of tRNA. The authors generated the model by negative stain EM on TAP-purified complexes from baker's yeast. The EM scaffold was estimated as 31Å and 27Å. To refine placements into the Holo structure, they use cross-linking mass spectrometry to determine relative subunit orientation in the structure. This allowed for the fit of recently published and newly generated structures leading to a full model. In addition to the modeling work the authors tested some of the interactions that they observed in the MS and predicted from the model by GST-pulldown. The new model also fits a tRNA that was modeled into a site previously described as critical for tubulin-acetylation. According to an indirect assay they find suggestive evidence that the mutations in the domain impair tRNA modification.

The holoElongator complex consists of 2x each subunit of Elp1-6. Older hypothetical models (also by the authors) had placed the ring of Elp456 in the middle of a putative holo-structure surrounded by two copies of Elp123. The current manuscript disproves the idea and shows that the holo-complex is asymmetrical and that a wing-shaped Elongator123 forms the structural basis of the holo-complex, where an Elp456 ring binds to one of the Elp123 subunits. Finally, the model may provide an explanation why the mutation prevalent in most patients of familial dysautonomia leads to a dysfunctional Elongator complex. This is a surprising finding that has implications on how the Elongator complex acts as a molecular machine. There are conflicting reports about different functions of the Elongator complex. Careful analysis of the structural determinants of the complex will ultimately solve these discrepancies. However, the authors are careful not to overstate their finding. The work is suited for a publication in EMBO Reports if some points are addressed.

1. Can the authors discuss why there are so few elp4, 5, 6 crosslinks?

Response: In general in XL-MS, low number of crosslinks can be observed for many reasons. Some reasons relate to the geometrical and physical properties of the proteins such as less well integration of subunits into the core complex, inaccessibility of lysine side chains for modification (e.g. if they are buried in an interaction interface), or unfavorable geometric orientation of the side chains. Other causes are related to mass spectrometry identification of the crosslinked peptides such as varying susceptibility to digestion, liquid chromatography, and ionization. In our experience, it is difficult to identify and disentangle these factors and several of them can influence crosslink numbers in the same time. In the specific case of Elongator, Elp1, Elp3, Elp4 have many regions predicted to be disordered and they lead to many crosslinks (Figure 2). In contrast, Elp5 and Elp6 have less disordered regions (the missing C-terminal region in Elp5 is predicted to be ordered) which we believe could be one reason for the lower number of crosslinks in Elp5 and Elp6. To provide an explanation to the interested reader we have rewritten the comment on page 9 to: "Comparing to Elp1, Elp3 and Elp4, we observed only very few inter-subunit crosslinks in Elp2, Elp5 and Elp6. This may result from the lower level of disordered regions in Elp2, Elp5, and Elp6, but also may suggest that these subunits simply share less and smaller interfaces with the other subunits."

2. The authors claim that "Elp1 interacts with KAT domain of Elp3, but shows only very weak interaction with the SAM domain of Elp3" (Figure 4A). This seems an overstatement as a band of Elp1 is seen in both pull-downs. Please Change.

We agree with the referee that we slightly overstated the finding. Nevertheless, the intensity of the Elp1 band interacting with Elp3-SAM is as strong as the background, which becomes stronger only in the presence of 456. In contrast, the interaction between the KAT domain and Elp1 appears robustly above the background also in the absence of Elp456. We have rephrased the sentence on page 12/13 accordingly – "Consistent with our crosslinking results and as predicted from our model, we observed that Elp1 interacts directly with the KAT domain of Elp3, but requires Elp456 to interact with the SAM domain of Elp3."

3. The authors claim that mutant Elongator leads to phenotypes typical for RNA modification defects. Please state clearly that this is not a proof of a modification defect, but rather an indirect assessment of tRNA modification. The main weakness is that the authors do not provide direct evidence for the absence of xm5 formation. But this may be beyond the scope of the article.

Response: We agree with the reviewer that we do not directly quantify or detect the level of modified tRNAs or xm5 modified wobble base positions. Therefore, we rephrased the affected sections and refer simply to the activity of Elongator. Nevertheless, we would like to mention that several studies have shown that most (if not all) phenotypes associated with Elongator's cellular activity can be rescued by the overexpression of certain tRNAs, indicating a more indirect involvement in other cellular functions, apart from tRNA modifications. We rephrased the respective sentences on page 14: "Furthermore, we also show that two additional regions (aa201-204 and aa552-557) seem to be important for Elongator activity as they show similar phenotypes (Fig. EV5A,B)." and on page 16 "Although we show that mutating the residues of this patch (R626, R628, R654, R675) affects Elongator's activity, it is not involved in protein-protein interactions within the Elp123 sub-complex (Fig. 6D)."

4. A general comment: The preparations of protein complexes do not appear completely clean or not well labelled in Figure 1A. However, the sample on the grid appears homogeneous. What are the additional bands and why don't they matter?

Response: All bands from the Elp6TAP purification (Fig 1A) were also analysed by mass spectrometry confirming that the three first bands correspond to Elp1. The bands of Elp2, Elp3 and Elp4 are also confirmed by MS. The contamination band below Elp4 (Mw marker, 50kDa) corresponds to a degradation of Elp4, as identified by MS. Finally, the bands for Elp5 and Elp6 have the same molecular weight (due to the remaining CBP/TEV tag), and have been also confirmed by MS. All bands from the Elp123 sample (Fig 2A) were also analysed by MS. The three first bands correspond to the Elp1 protein, and several phosphorylated peptides of Elp1 were identified in these three bands. Again, Elp2 and Elp3 were confirmed by MS. The light bands in Elp123 may represent leftovers of Elp456, and Elp4 has been identified by MS. Elp5 and Elp6 are much more difficult to detect by MS due to its small molecular weight and the low number of available trypsin sites. On top of that, the gels in Fig. 1A and 2A show the samples prior to the gel filtration step, so presumably the samples are further purified after the GF step, as we observed with non-cross linked sample. However, as the samples used in this study were cross-linked before the GF step, no separate bands can be detected on a gel at the stage of grid preparation.

2nd Editorial Decision

08 November 2016

Thank you for the submission of your revised manuscript. It was sent back to referee 1, who supports its publication now. The referee only has two minor suggestions that I would like you to address in the manuscript text before we can proceed with the official acceptance. If only text changes are made, you can send us the new text file by email, and we will replace the current file for you. I look forward to seeing a final version of your manuscript as soon as possible.

REFEREE REPORTS

Referee 1:

The authors have made several improvements in their revised manuscript, and now provide tilt pair validation, additional crosslinking information, and have made small changes in data presentation and the main text. The main finding of this work remains the unexpected asymmetry in the Elongator structure, and the resultant architectural model. In the future, it will be of great interest to learn how this asymmetry relates to Elongator function. Although not necessary for publication of the current work, the authors may still wish to additionally validate their subunit assignments using a larger protein tag than GFP, as detailed in the authors responses. For example, by MBP-fusion or Antibody-labelling of N- or C-termini of Elp1. All the data presented here is of high quality and entirely consistent with the proposed architectural model, and in my opinion, this work should be published in its current form.

Minor points:

1. Consistent with previous data, the authors suggest that the multiple bands from Elongator and Elp123 purifications derive at least in part from different phosphorylation states (see Fig. 1A, 2A). This could be mentioned in the main text or in figure legends of Figure 1 and 2 figure for clarity, since the additional protein bands are very noticeable and the authors write that their sample is of great purity. The Elp1 phosphorylation state may further affect Elongator function and complex conformation and rigidity. Together with such a statement the authors could point to the relevant literature, such as Abdel-Fattah et al. 2015, Plos Genetics (reference 34 in this manuscript).

2. To my knowledge Setiapura et al. do not propose that two Elp456 rings bind transiently to Elp123 at the same time, neither do they provide evidence for this. The statement in the main text should therefore be removed: "Though we cannot exclude the possibility of two Elp456 rings bound to the Elp123 sub-complex in a transient state as suggested by Setiapura et al. [42] [...]".

2nd Revision - authors' response

08 November 2016

The authors made the requested changes and submitted updated files by email.

3rd Editorial Decision - Acceptance

14 November 2016

I am very pleased to accept your manuscript for publication in the next available issue of EMBO reports. Thank you for your contribution to our journal.

YOU MUST COMPLETE ALL CELLS WITH A PINK BACKGROUND ↓
PLEASE NOTE THAT THIS CHECKLIST WILL BE PUBLISHED ALONGSIDE YOUR PAPER

Corresponding Author Name: Sebastian Glatt & Christoph W Mueller
Journal Submitted to: EMBO Reports
Manuscript Number: EMBOOR-2016-43353-7